# Engineering active sites on hierarchical transition bimetal oxides/sulfides heterostructure array enabling robust overall water splitting

Panlong Zhai[1,5], Yanxue Zhang[2,5], Yunzhen Wu[1,5], Junfeng Gao[2], Bo Zhang[1], Shuyan Cao[1], Yanting Zhang[1], Zhuwei Li[1], Licheng Sun [1,3,4] & Jungang Hou[1✉]

Rational design of the catalysts is impressive for sustainable energy conversion. However, there is a grand challenge to engineer active sites at the interface. Herein, hierarchical transition bimetal oxides/sulfides heterostructure arrays interacting two-dimensional $MoO_x$/$MoS_2$ nanosheets attached to one-dimensional $NiO_x$/$Ni_3S_2$ nanorods were fabricated by oxidation/hydrogenation-induced surface reconfiguration strategy. The $NiMoO_x$/NiMoS heterostructure array exhibits the overpotentials of 38 mV for hydrogen evolution and 186 mV for oxygen evolution at 10 mA cm$^{-2}$, even surviving at a large current density of 500 mA cm$^{-2}$ with long-term stability. Due to optimized adsorption energies and accelerated water splitting kinetics by theory calculations, the assembled two-electrode cell delivers the industrially relevant current densities of 500 and 1000 mA cm$^{-2}$ at record low cell voltages of 1.60 and 1.66 V with excellent durability. This research provides a promising avenue to enhance the electrocatalytic performance of the catalysts by engineering interfacial active sites toward large-scale water splitting.

[1] State Key Laboratory of Fine Chemicals, School of Chemical Engineering, Dalian University of Technology, 116024 Dalian, P. R. China. [2] Laboratory of Materials Modification by Laser, Ion and Electron Beams, Dalian University of Technology, Ministry of Education, 116024 Dalian, P. R. China. [3] College of Science, Westlake University, 310024 Hangzhou, P. R. China. [4] Department of Chemistry, School of Engineering Sciences in Chemistry, Biotechnology and Health, KTH Royal Institute of Technology, 10044 Stockholm, Sweden. [5] These authors contributed equally: Panlong Zhai, Yanxue Zhang, Yunzhen Wu. ✉email: jhou@dlut.edu.cn

The generation of clean energy from water electrolysis is a feasible solution to overcome the problems of energy issues[1]. The sustainable alternative for hydrogen generation is electrocatalytic water splitting, involving hydrogen evolution reaction (HER) and oxygen evolution reaction (OER)[2]. Generally, noble materials, Pt for HER and $RuO_2$ or $IrO_2$ for OER, are typical electrocatalysts. Nevertheless, the practical application is limited by the use of noble materials owing to the scarcity and the high cost. To this end, it is interesting to produce bifunctional materials by the integration of OER and HER catalysts towards water splitting in various media[3]. To address the challenges, catalyzing HER, OER and overall water splitting have been conducted by extensive catalysts, such as oxides, hydroxides, phosphides, nitrides and chalcogenides[4–11]. Thus, it is urgently needed to design earth-abundant and low-cost non-noble-metal catalysts for industrial applications.

Among various materials, Mo- and Ni-based sulfides are promising transition-metal electrocatalysts. To improve the performance of these catalysts, various strategies, such as morphology engineering, defect engineering, and heterostructure engineering have been adopted in this field. Architectural nanostructures have been controlled by the synthesis regulation of the electrocatalysts owe to inherent anisotropy and high flexibility[5–8]. Inspired by the advantages of the architectures, the integration of different nanostructures can effectively optimize the electrocatalytic performance. For example, $MoO_3$ nanodots supported on $MoS_2$ monolayer, $MoNi_4$ anchored $MoO_2$ cuboids or $MoO_{3-x}$ nanorods and $NiS_2$/N-$NiMoO_4$ nanosheets/nanowires have been produced for the excellent electrocatalytic water splitting[12–15], providing an appealing platform with the hierarchical nanostructures. Apart from morphology engineering, the hybrids can be extensively constructed by use of different transition-metal electrocatalysts through heterostructure engineering, regulating electron transfer and active site as well as the activity owe to the construction of coupling interfaces and the synergistic effect of the heterostructures. For instance, a large number of the heterostructures, such as $NiMo$/$NiMoO_x$[8], $Co_3O_4$/$Fe_{0.33}Co_{0.66}P$[16], $Ni_2P$/$NiP_2$[17], $NiFe(OH)_x$/$FeS$[18], $Pt_2W$/$WO_3$[19], $CuCo$/$CuCoO_x$[20], $Co(OH)_2$/PANI[21], $FeOOH$/Co/$FeOOH$[22], $Co_{0.85}Se$/NiFe/graphene[23], $Ni_3N$/VN[24], NiCu–NiCuN[25], have been extensively synthesized for the enhanced electrochemical activities. Typically, sulfides-based heterostructures, such as CoS-doped $\beta$-$Co(OH)_2$/$MoS_{2+x}$[26], $MoS_2$/$Fe_5Ni_4S_8$[27], $MoS_2$/$Ni_3S_2$[28], $NiS_2$/$MoS_2$[29], $MoS_2$/$Co_9S_8$/$Ni_3S_2$/Ni[30], and $MoS_2$/$(Co,Fe,Ni)_9S_8$ coupled FeCoNi-based arrays[31], have been systematically explored for the improved activities of electrochemical water splitting. With regard to transition-metal dichalcogenides, $MoS_2$ and $Ni_3S_2$ materials have been substantially explored as HER electrocatalysts[32–35]. However, the HER performance of transition metal sulfides is limited by poor charge transport, low active site reactivity, and inefficient electrical contact with the supported catalysts[36]. Especially, the generation of S–$H_{ads}$ bonds (H atoms adsorption, $H_{ads}$) on the surface of metal sulfides is beneficial for H adsorption, while it is difficult to conduct the conversion of the $H_{ads}$ to $H_2$[36,37]. However, the OER performance of metal sulfides remains far from satisfactory[27–31]. Owe to long-time durability as major obstacle, there is less report about the electrocatalysts, delivering large catalytic current densities (e.g., 500 and 1000 mA cm$^{-2}$) for practical application[38–41]. Based on the above-mentioned analysis, it is essential to design the rational heterostructures through the combined regulation of architectural morphology and heterostructures, engineering active sites, optimizing energy adsorption, and accelerating water splitting kinetics towards large-scale electrolysis.

Herein, three-dimensional (3D) $NiMoO_x$/NiMoS heterostructure array is fabricated by surface reconfiguration strategy through oxygen plasma as oxidation treatment and subsequent hydrogenation regulation by use of NiMoS architecture as the precursor, interacting two-dimensional (2D) $MoO_x$/$MoS_2$ nanosheets attached to one-dimensional (1D) $NiO_x$/$Ni_3S_2$ nanorods array. As-synthesized $NiMoO_x$/NiMoS array presents the remarkable electrocatalytic performance, achieving the low overpotentials of 38, 89, 174, and 236 mV for HER and 186, 225, 278, and 334 mV for OER at 10, 100, 500, and 1000 mA cm$^{-2}$, even surviving at large current densities of 100 and 500 mA cm$^{-2}$ with long-term stability. The remarkable electrocatalytic performance of transition bimetal oxides/sulfides heterostructure array as the industrially promising electrocatalyst is ascribed to not only the simultaneous modulation of component and geometric structure, but also the systematic optimization of charge transfer, abundant electrocatalytic active sites, and exceptionally synergistic effect of the heterostructure interfaces. The turnover frequency (TOF) of $NiMoO_x$/NiMoS array at the overpotential of 100 mV is ~45 times higher than that of NiMoS array. Density functional theory calculations reveal that the coupling interface between $NiMoO_x$ and NiMoS optimizes adsorption energies and accelerates water splitting kinetics, thus promoting the electrocatalytic performance. Especially, the assembled two-electrode cell by use of $NiMoO_x$/NiMoS array delivers the industrially required current densities of 500 and 1000 mA cm$^{-2}$ at the low cell voltages of 1.60 and 1.66 V, along with excellent durability, thus holding great promise for industrial water splitting application.

## Results

**Synthesis and characterization.** The hierarchical $NiMoO_x$/NiMoS array was fabricated by oxidation/hydrogenation-induced surface reconfiguration strategy by use of NiMoS precursor, assembling as two-electrode cell towards industrially electrocatalytic water splitting (Fig. 1 and Supplementary Fig. 1). To determine the crystal structure, X-ray diffraction (XRD) patterns of NiMoS-based arrays are showed (Supplementary Fig. 2). Based on the hydrothermal reaction, the representative peaks of the precursors can be assigned to the planes of $MoS_2$ phase (JCPDS No. 37-1492) and $Ni_3S_2$ phase (JCPDS No. 44-1418), confirming the formation of individual $MoS_2$ and $Ni_3S_2$ as well as $MoS_2$/$Ni_3S_2$ heterostructure as the precursors (Supplementary Fig. 2a). After oxygen plasma as oxidation treatment and subsequent hydrogenation regulation, several $MoO_3$ (JCPDS No.47-1320), $MoO_2$ (JCPDS No. 50-0739), and NiO (JCPDS No.44-1159) phases as well as the mixed $MoO_3$/$MoO_2$/NiO/Ni phases are observed in $MoS_2$, $Ni_3S_2$, and $MoS_2$/$Ni_3S_2$ (Supplementary Fig. 2b). Thus, all above-mentioned results demonstrate the successful formation of $NiO_x$/$Ni_3S_2$, $MoO_x$/$MoS_2$, and $NiMoO_x$/NiMoS heterostructure arrays.

To confirm the geometric morphologies of individual arrays by scanning electron microscope (SEM), as shown in Fig. 2a, d, $MoS_2$ nanosheets as the precursor with an average size over 1 μm are homogeneously supported on the conductive substrate. While the rough surface of $Ni_3S_2$ array as the precursor is observed (Supplementary Fig. 3). Interestingly, two-dimensional (2D) $MoS_2$ nanosheets with an average size below 1 μm are attached to one-dimensional (1D) $Ni_3S_2$ nanorods array on 3D foam substrate, resulting into the formation of hierarchical $MoS_2$/$Ni_3S_2$ (denoted as NiMoS) heterostructure array (Fig. 2b, e). After the oxidation/hydrogenation treatment of NiMoS array, there is no obvious change upon the main morphology for 3D $NiMoO_x$/NiMoS heterostructure array. However, the small size and rough surface of $MoS_2$ nanosheets in $NiMoO_x$/NiMoS array are observed in comparison of $MoS_2$ in NiMoS array (Fig. 2cf). Meanwhile, the energy-dispersive X-ray (EDX) spectra and

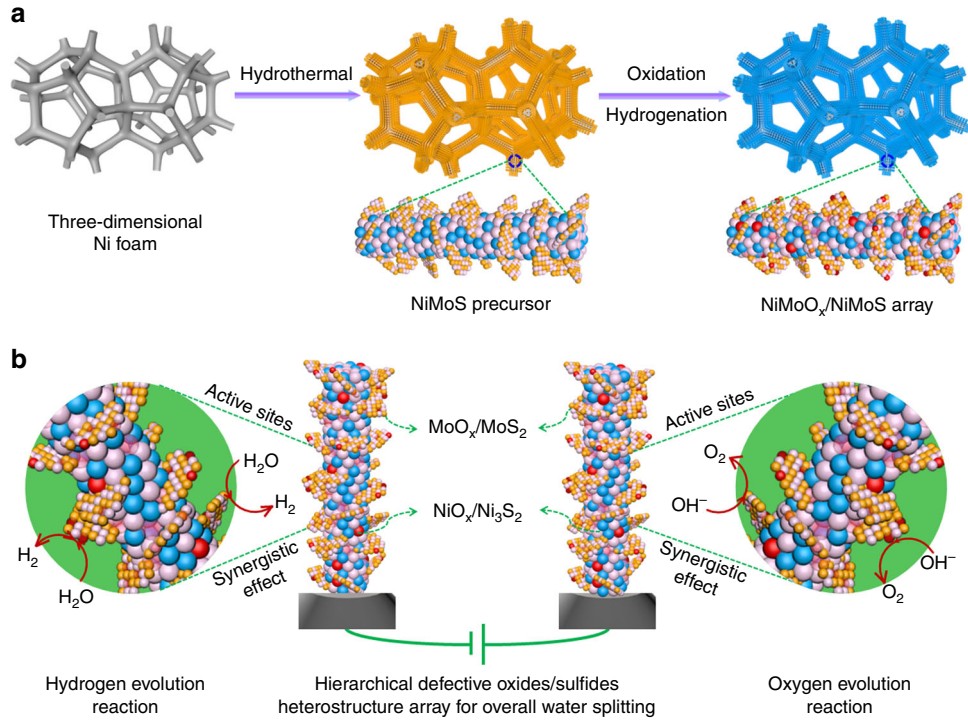

**Fig. 1 Schematic representation of synthesis and overall water splitting. a** Synthesis illustration of transition bimetal oxides/sulfides heterostructure array. **b** NiMoO$_x$/NiMoS array as two-electrode-cell towards large-scale electrolysis. Colored balls represent various elements (blue: Mo, pink: S, red: O, yellow: Ni).

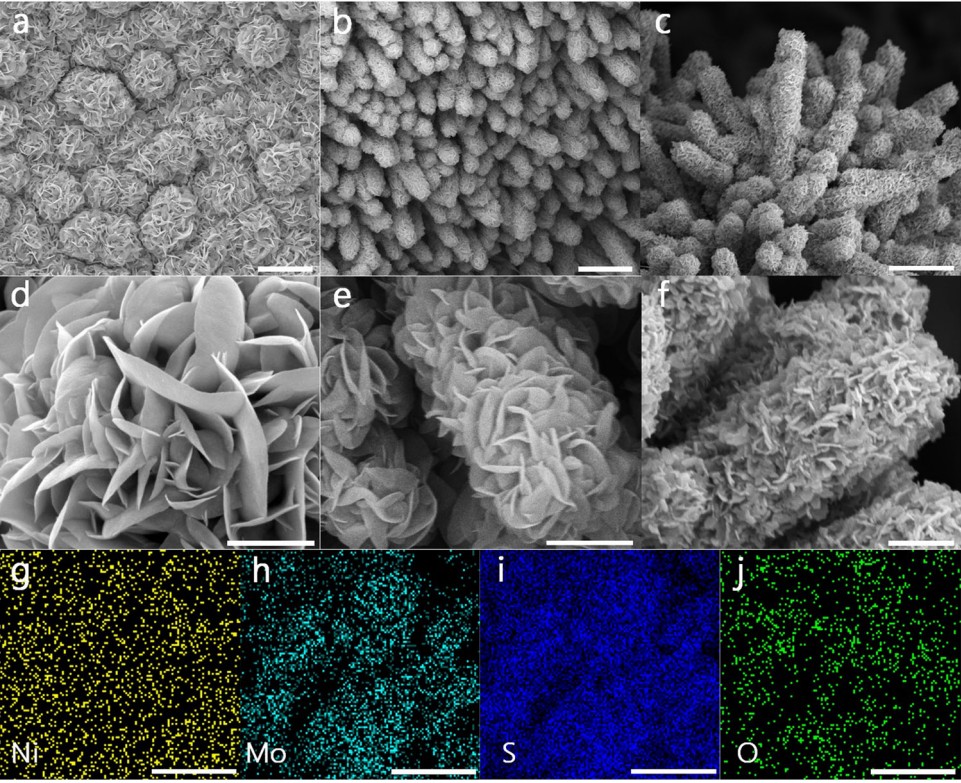

**Fig. 2 Morphological and structural characterizations.** SEM images of **a**, **d** MoS$_2$, **b**, **e** NiMoS, **c**, **f** NiMoO$_x$/NiMoS. **g–j** Elemental mapping images of NiMoO$_x$/NiMoS. Scale bar, **a–c** 5 μm; **d–f** 1 μm; **g–j** 10 μm.

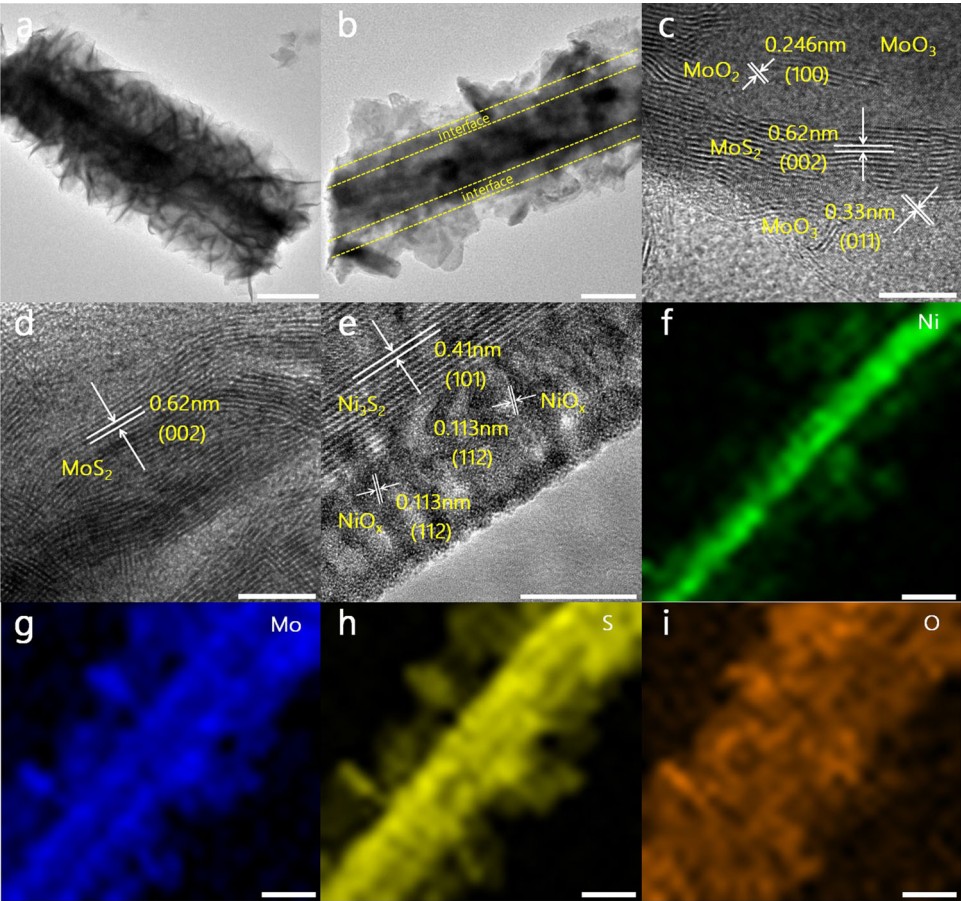

**Fig. 3 Morphological and structural characterizations.** TEM and HRTEM images of **a**, **d** NiMoS and **b**, **c**, **e** NiMoO$_x$/NiMoS. **f–i** Elemental distribution mapping of Ni, Mo, S, and O in NiMoO$_x$/NiMoS. Scale bar, **a**, **b** 500 nm; **c–e** 5 nm; **f–i** 200 nm.

elemental mapping (Fig. 2 and Supplementary Fig. 4) indicate the molar content of MoO$_x$/MoS$_2$ about 6.1% and the homogeneous element distribution in NiMoO$_x$/NiMoS array. Thus, the above-mentioned analysis indicates the formation of NiMoO$_x$/NiMoS array as 3D integrated architectures.

To check the details of the morphology, transition electron microscope (TEM) and high-resolution TEM observations verify the architectures of NiMoS and NiMoO$_x$/NiMoS heterostructure arrays, indicating that MoS$_2$ and MoO$_x$/MoS$_2$ nanosheets are attached to Ni$_3$S$_2$ and NiO$_x$/Ni$_3$S$_2$ nanorods arrays, respectively (Fig. 3). Compared to MoS$_2$ and Ni$_3$S$_2$ in NiMoS nanostructures, the characteristic lattice fringes of 0.62, 0.33, and 0.246 nm can be assigned to the (002) plane of MoS$_2$, (011) plane of MoO$_3$, and (100) plane of MoO$_2$ and even more, the (101) plane of Ni$_3$S$_2$ and the (112) plane of NiO$_x$ can be proven by the lattice fringes of 0.41 and 0.113 nm in NiMoO$_x$/NiMoS heterostructures. Typically, the arrangements of MoO$_x$ and NiO$_x$ layers are observed on the surface of MoS$_2$ and Ni$_3$S$_2$, indicating the formation of NiMoO$_x$/NiMoS heterostructure array. Moreover, the elemental mappings by high-angle annular dark-field scanning transmission electron microscopy (HAADF-STEM) confirm the uniform distribution of Ni, Mo, S, and O (Fig. 3 and Supplementary Fig. 5). Therefore, the whole results of SEM and TEM analysis confirm the formation of 3D NiMoO$_x$/NiMoS heterostructure array as the integrated architectures.

To conduct the chemical valences of the heterostructures, X-ray photoelectron spectroscopy (XPS) spectrum has been tested in Fig. 4. With regard to Mo 3$d$ regions, the main peak could be split into two distinct peaks of Mo 3$d_{5/2}$ (229.1 eV) and Mo 3$d_{3/2}$ (232.4 eV), indicating the dominance of Mo$^{4+}$ in NiMoS

(Supplementary Fig. 6)[8,42]. The peaks at 855.2, 861.5, 872.9 and 879.5 eV can be indexed to Ni 2$p_{3/2}$ and Ni 2$p_{1/2}$ orbitals as well as two satellites in NiMoS (Supplementary Fig. 6)[25]. However, the signals at 229.3, 232.4, and 235.5 eV can be indexed to Mo$^{4+}$ 3$d_{5/2}$, Mo$^{4+/6+}$ 3$d_{3/2}$, and Mo$^{6+}$ 3$d_{3/2}$ orbitals, confirming the existence of Mo$^{4+}$ and Mo$^{6+}$ in NiMoO$_x$/NiMoS owe to the formation of MoO$_x$[26]. For Ni 2p orbitals, there is a shift upon the peak positions and the two new peaks at 854.6 and 852.6 eV, demonstrating the existence of Ni–O bonds and metallic Ni$^0$ and the formation of NiO$_x$ species in NiMoO$_x$/NiMoS[8,25]. Typically, the signals at 529.5 and 531.5 eV for O 1s belong to typical metal-oxygen bonds and oxygen vacancies in NiMoO$_x$/NiMoS heterostructure[8]. With regard to S 2$p$ peaks, the negative shift is observed in NiMoO$_x$/NiMoS with the increasing temperature of thermal treatment, demonstrating the loss of S and the formation of S vacancies[43]. The similar phenomenon of O 1s and S 2$p$ is observed in MoO$_x$/MoS$_2$ and NiO$_x$/Ni$_3$S$_2$ heterostructures (Supplementary Fig. 7–8). Thus, the combined analysis demonstrates the successful synthesis of hierarchical transition bimetal oxides/sulfides heterostructure array.

**Electrocatalytic HER performance.** The electrocatalytic performance of various arrays in the three-electrode system was conducted through a linear scan voltammogram (LSV) in 1 M KOH solution at 25 °C. The polarization curves of NiMoO$_x$/NiMoS, MoO$_x$/MoS$_2$, NiO$_x$/Ni$_3$S$_2$, and NiMoS heterostructure arrays are presented in Fig. 5a, together with commercial Pt/C and Ni foam (Supplementary Fig. 9). In comparison of NiMoS (219, 392, and 611 mV), MoO$_x$/MoS$_2$ (163, 282, and 430 mV), NiO$_x$/Ni$_3$S$_2$ (67,

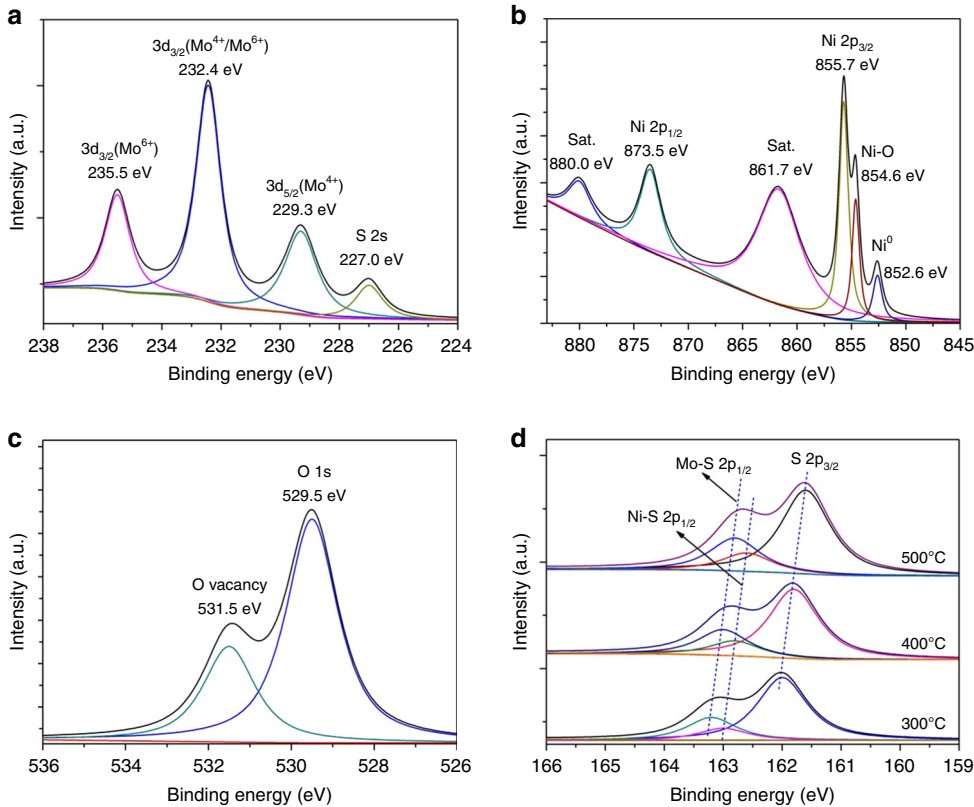

**Fig. 4 XPS spectra of NiMoO$_x$/NiMoS.** High-resolution XPS signals of **a** Mo 3*d*, **b** Ni 2*p*, **c** O 1s, **d** S 2*p* of NiMoO$_x$/NiMoS array with different thermal treatment temperatures.

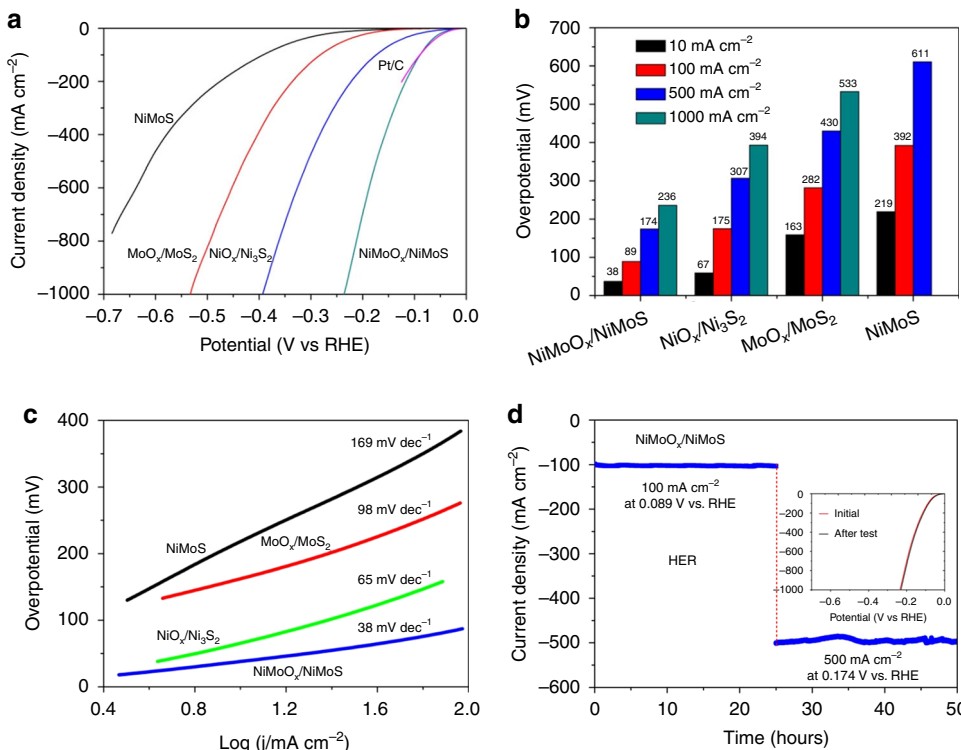

**Fig. 5 HER catalytic performance. a** HER polarization curves, **b** overpotentials at typical current densities, **c** Tafel slopes of NiMoS, MoO$_x$/MoS$_2$, NiO$_x$/Ni$_3$S$_2$, and NiMoO$_x$/NiMoS. **d** Time-dependent current density curves of NiMoO$_x$/NiMoS at typical potentials. Inset: polarization curves of NiMoO$_x$/NiMoS for the stability test.

175, and 307 mV), NiMoO$_x$/NiMoS array delivers the current densities of 10, 100, and 500 mA cm$^{-2}$ at the low overpotentials of 38, 89 and 174 mV, respectively, even requiring a low overpotential of 236 mV at a large current density of 1000 mA cm$^{-2}$ towards HER (Fig. 5b). It is worth mentioning that NiMoO$_x$/NiMoS array could surpass commercial Pt/C catalyst at the high overpotentials while comparable HER activity at the low potentials. Compared to most reported HER catalysts (Supplementary Table 1), the overpotential of NiMoO$_x$/NiMoS array at a current density of 10 mA cm$^{-2}$ is smaller than those of MoS$_2$ (170 mV)[44], CoFeZr oxides (104 mV)[6], CoS-Co(OH)$_2$@MoS$_{2+x}$ (140 mV)[26], MoS$_2$/Fe$_5$Ni$_4$S$_8$ (120 mV)[27], MoS$_2$/Ni$_3$S$_2$ (110 mV)[28], MoS$_2$/Co$_9$S$_8$/Ni$_3$S$_2$ (113 mV)[30], and O-CoMoS (97 mV)[42], etc. To regulate the capacities of charge transfer and active sites of NiMoS, it is interesting to determine the precise condition of plasma oxidation and hydrogenation treatment (Supplementary Fig. 10), indicating the best oxygen plasma power of 100 W and appropriate hydrogenation temperature of 400 °C of NiMoO$_x$/NiMoS array. To conduct HER kinetic mechanism, the lowest Tafel slope (Fig. 5c), 38 mV per decade of NiMoO$_x$/NiMoS array is obtained in comparison of NiO$_x$/Ni$_3$S$_2$ (65 mV dec$^{-1}$), MoO$_x$/MoS$_2$ (98 mV dec$^{-1}$), NiMoS (169 mV dec$^{-1}$), indicating the rapid HER kinetics of NiMoO$_x$/NiMoS array owe to the advantages of the construction of 3D heterostructured architectures and the introduction of the defects. After the analysis of electrochemical impedance spectroscopy (EIS), the lowest charge transfer resistance of NiMoO$_x$/NiMoS array due to the generation of defective species and metallic Ni is obtained in comparison of NiO$_x$/Ni$_3$S$_2$, MoO$_x$/MoS$_2$, and NiMoS (Supplementary Fig. 10). To explore the intrinsic electrocatalytic performance of each active sites, the turnover frequency (TOF) is calculated (Supplementary Table 2–3). The TOF value of NiMoO$_x$/NiMoS array (1.97 s$^{-1}$) at the overpotential of 100 mV is ~45 times higher than that of NiMoS array (0.0435 s$^{-1}$). Moreover, mass activity, 436 A g$^{-1}$ of NiMoO$_x$/NiMoS array is calculated at the overpotential of 200 mV (Supplementary Fig. 11), which is better than other non-nobel metal electrocatalysts (Supplementary Table 4). Generally, the electrochemically active surface area (ECSA) is regarded as an estimation of active sites and is proportional to the double-layer capacitance ($C_{dl}$)[45–47]. The highest $C_{dl}$ values of NiMoO$_x$/NiMoS array among all catalysts implies the maximum electroactive area (Supplementary Fig. 12). Moreover, the current of NiMoO$_x$/NiMoS and commercial Pt/C supported on Ni plate was normalized to ECSA (Supplementary Figs. 13–16), demonstrating a higher instrinsic activity of NiMoO$_x$/NiMoS catalyst in comparison of commercial Pt/C. Owe to the stability as pivot criterion for practical application, the time-dependent current density curves confirm that there is no obvious change upon the current densities of 100 and 500 mA cm$^{-2}$ at 0.089 and 0.174 V vs. RHE over 50 h (Fig. 5d). Afterwards, the amount of hydrogen evolution of NiMoO$_x$/NiMoS array is measured in comparison of theoretical quantity (Supplementary Fig. 17), presenting a promising Faradaic efficiency of 99.6 ± 0.3% towards real water splitting into hydrogen. Based on the above-mentioned analysis, the synergistic action of morphology and heterostructure engineering upon NiMoO$_x$/NiMoS array can modulate the unique architectures, optimize the charge transfer and catalytic active sites, and thus improve HER performance.

**Electrocatalytic OER performance**. In general, the efficiency is always limited by OER as major barrier for overall water splitting. In our system, NiMoO$_x$/NiMoS array exhibits the best OER performance among all arrays, together with commercial IrO$_2$ catalyst and Ni foam (Fig. 6a and Supplementary Fig. 18). In comparison of NiMoS (370, 437, and 526 mV), MoO$_x$/MoS$_2$ (266, 332, and

438 mV), NiO$_x$/Ni$_3$S$_2$ (214, 267, and 366 mV), as-synthesized NiMoO$_x$/NiMoS array presents the low overpotentials of 186, 225, and 278 mV at current densities of 10, 100, and 500 mA cm$^{-2}$, and delivers a large current density of 1000 mA cm$^{-2}$ at 334 mV towards OER (Fig. 6b), satisfying the requirements for commercial electrocatalytic application (for example, $j \geq 500$ mA cm$^{-2}$ at $\eta \leq 300$ mV)[48–51]. Compared to most reported OER catalysts (Supplementary Table 5), the overpotential of NiMoO$_x$/NiMoS array at 10 mA cm$^{-2}$ is still lower than those of O-CoMoS (272 mV)[42], CoS-Co(OH)$_2$@MoS$_{2+x}$ (380 mV)[26], MoS$_2$/Fe$_5$Ni$_4$S$_8$ (204 mV)[27], MoS$_2$/Ni$_3$S$_2$ (218 mV)[28], and iron-substrate-derived electrocatalyst (269 mV)[48], etc. Especially, the influence of oxygen plasma power and hydrogenation temperature upon the OER performance of NiMoO$_x$/NiMoS array is determined (Supplementary Fig. 19), confirming the best plasma power of 100 W and thermal treatment temperature at 400 °C of NiMoO$_x$/NiMoS array. To in-depth understand the OER kinetic mechanism, the lowest Tafel slope, 34 mV per decade of NiMoO$_x$/NiMoS is achieved in comparison of NiO$_x$/Ni$_3$S$_2$ (56 mV dec$^{-1}$), MoO$_x$/MoS$_2$ (62 mV dec$^{-1}$), NiMoS (74 mV dec$^{-1}$), demonstrating the fast OER kinetics of NiMoO$_x$/NiMoS (Fig. 6c). Remarkably, the largest $C_{dl}$ value of 21.5 mF cm$^{-2}$ of NiMoO$_x$/NiMoS is obtained by the evaluation of ECSA among all arrays (Supplementary Fig. 20), indicating the production of abundant active sites in NiMoO$_x$/NiMoS array. Especially, the high ECSA of NiMoO$_x$/NiMoS array confirms the advantages of the exposed component and geometric structures of sufficient electrocatalytic active sites. Interestingly, the high-valence Mo and Ni species are obtained in NiMoO$_x$/NiMoS array during the OER process (Supplementary Fig. 24), indicating the possible generation of hydroxyl oxides as the actual surface active sites and thus enhancing the OER activities owe to the synergistic action of 3D architectures and the heterostructures. In particular, NiMoO$_x$/NiMoS array can preserve OER activities at 100 and 500 mA cm$^{-2}$ with the potentials of 1.455 and 1.508 V vs. RHE over 50 h (Fig. 6d), indicating the fascinating OER stability. Typically, the amount of oxygen evolution of NiMoO$_x$/NiMoS array is measured in comparison of theoretical quantity (Supplementary Fig. 21), presenting OER Faradaic efficiency of 97.5 ± 0.4% owe to the synergistic effect of the morphology and heterostructure engineering.

**Electrocatalytic performance for overall water splitting**. Inspired by excellent HER and OER performance, NiMoO$_x$/NiMoS array was assembled as cathode and anode in the two-electrode system. Impressively, the robust catalytic performance is achieved by as-synthesized NiMoO$_x$/NiMoS||NiMoO$_x$/NiMoS electrode (Fig. 7a), requiring the low cell voltages of 1.46, 1.62, 1.75, and 1.82 V at 10, 100, 500, and 1000 mA cm$^{-2}$ in 1 M KOH at 25 °C. In comparison of Ni–Fe–MoN[52], Fe$_{0.09}$Co$_{0.13}$-NiSe$_2$[53], NC/CoCu/CoCuO$_x$[20], MoS$_2$/Co$_9$S$_8$/Ni$_3$S$_2$[30], Pt-CoS$_2$[47], NC/NiCu/NiCuN[25], NC/NiMo/NiMoO$_x$[8], MoS$_2$/NiS$_2$[54], O-CoMoS[42], N-NiMoO$_4$/NiS$_2$[15], MoS$_2$/NiS[55], P-Co$_3$O$_4$[56], Ni/Mo$_2$C[57], CoNi(OH)$_x$ | NiN$_x$[58], NiCo$_2$S$_4$[59], FeOOH[60], Ni$_5$P$_4$[61], NiCo/NiCoO$_x$[62], Fe-Ni@NC-CNT[63], Co$_x$PO$_4$/CoP[64], and commercial Pt/C||IrO$_2$ electrodes (Fig. 7c and Supplementary Table 6), the lower voltage at 10 mA cm$^{-2}$ is obtained for NiMoO$_x$/NiMoS array. Owe to excellent electrocatalytic performance, the two-electrode cell can also be evaluated by a 1.5 V AAA battery (Supplementary Fig. 22). Based on the analysis of the superaerophobicity by bubble contact tests (Supplementary Fig. 23), the superior bubble contact angle, 151.2° of NiMoO$_x$/NiMoS is obtained, demonstrating that this typical architecture could facilitate the release of the evolved gas bubbles and thus avoid the block of the catalyst active site. To be interesting, the hydrogen and oxygen bubbles

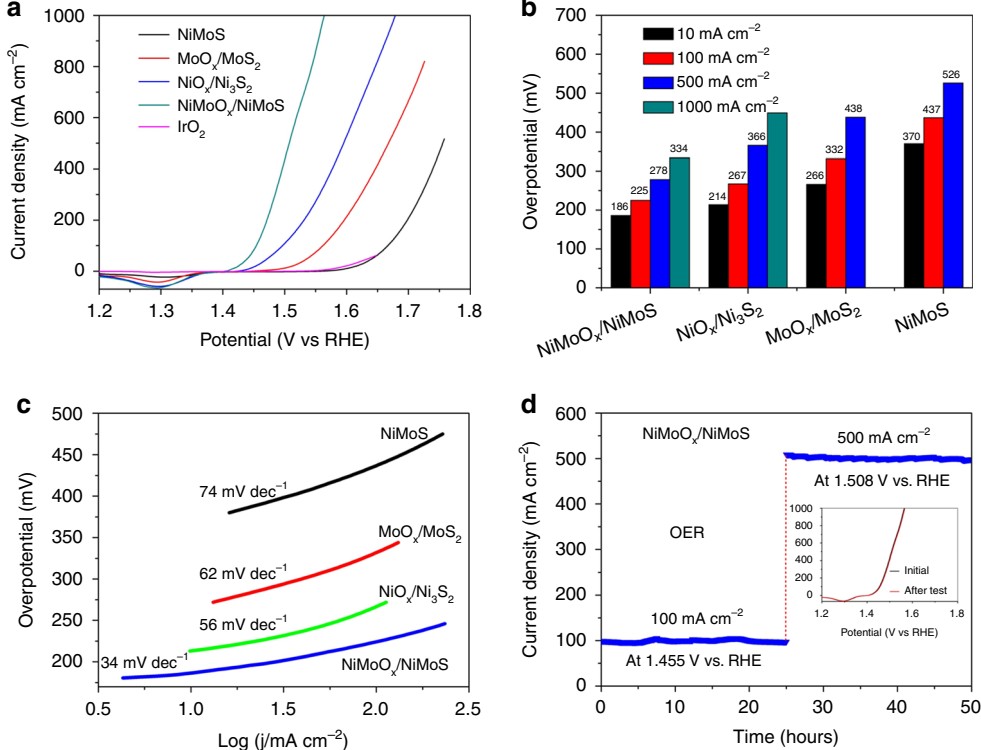

**Fig. 6 OER catalytic performance. a** OER polarization curves, **b** overpotentials at typical current densities, **c** Tafel slopes of NiMoS, $MoO_x/MoS_2$, $NiO_x/Ni_3S_2$, and $NiMoO_x/NiMoS$. **d** Time-dependent current density curves of $NiMoO_x/NiMoS$ at typical potentials. Inset: polarization curves of $NiMoO_x/NiMoS$ for the durability test.

escape effectively from the surface of $NiMoO_x/NiMoS$ array (Supplementary Movie). Moreover, the industrial environment is employed to explore the potential for industrialization applications. Typically, the record low voltages of 1.60 and 1.66 V of the two-electrode system in 6 M KOH solution at 60 °C are achieved for the industrial current densities of 500 and 1000 mA cm$^{-2}$, respectively, and it is still better than that of Pt/C‖IrO$_2$ couple (Fig. 7b and Supplementary Tabel 7). Compared to the reported electrocatalysts with the large current densities (e.g., 500 and 1000 mA cm$^{-2}$), such as NiMoN@NiFeN[65], nickel-cobalt complexes hybridized MoS$_2$[66], Ni-P-B/paper[49], NiVIr-LDH ‖NiVRu-LDH[50], phosphorus-doped Fe$_3$O$_4$[51], graphdiyne-sandwiched layered double-hydroxide nanosheets[67], N,S-coordinated Ir nanoclusters embedded on N,S-doped graphene[68], Co$_3$Mo/Cu[69], and FeP/Ni$_2$P hybrid[70], all aforementioned analysis confirm that as-prepared $NiMoO_x/NiMoS$ array could be served as promising industrial candidate for overall water splitting. With regard to the operating stability as important metric, this typical two-electrode cell can maintain the excellent electrocatalytic activity at a large current density of 500 mA cm$^{-2}$ at the voltage of 1.75 V over 500 h without obvious degradation in 1 M KOH solution at 25 °C (Fig. 7d). After HER, there is no obvious change upon the binding energies of various metal ions (Supplementary Fig. 24). However, the positive shift of two peaks located at 856.3 and 874.1 eV is observed in the XPS of Ni 2p, demonstrating that the oxidation of Ni$^{2+}$ to high valence state of Ni$^{3+}$, alone with the existence of new peak at 869.05 eV (Supplementary Fig. 24), thus indicating the formation of hydroxides and oxyhydroxides as the real active sites during OER process[30–34]. Although the hydroxides and oxyhydroxides are formed on the surface of $NiMoO_x/NiMoS$ array, there is no apparent change upon the morphology of the heterostructures (Supplementary Fig. 25), indicating the superior stability. Based on the above analysis, it is proven that $NiMoO_x/$

NiMoS array is excellent and stable system for overall water splitting, presenting the industrial hope.

**First-principles calculations.** To explore the original relationship between the intrinsically catalytic activity and the electronic and atomic structures of the interface of $NiMoO_x/NiMoS$, density functional theory calculations were performed to conduct the Gibbs free energies of every step in HER and OER (Supplementary Fig. 26–33). The hydrogen absorption energy ($\Delta G_{H*}$) is generally considered as the key descriptor for evaluating the performance of HER[71]. The sulfur sites of $NiO_x/Ni_3S_2$ and $MoO_x/MoS_2$ exhibit much lower $\Delta G_{H*}$ relative to that of Ni$_3$S$_2$ and MoS$_2$ (Fig. 8a and Supplementary Fig. 29–32), indicating that the integration of the oxides and sulfides enables the favorable H* adsorption and the tremendous decrement of thermodynamic barriers for hydrogen production. Especially, the oxidation/hydrogenation-induced surface reconfiguration results into the fabrication of $NiMoO_x/NiMoS$ heterostructure. The sulfur species serve as the distinctive active sites for the optimized hydrogen adsorption with nearly zero $\Delta G_{H*}$ (0.003 eV), in comparison of $NiO_x/Ni_3S_2$ ($\Delta G_{H*} = 0.074$ eV) and $MoO_x/MoS_2$ ($\Delta G_{H*} = 0.422$ eV). Since oxygen-free NiMoS shows much more negative $\Delta G_{H*}$ (−0.284 eV) comparing to $NiMoO_x/NiMoS$, it is hypothesized that the oxide species of the unique multi-interfaces may avoid the excessively strong adsorption of H* and bring about the facile intermediates desorption. Theoretically, water oxidation in alkaline medium involves four concerted proton-electron transfer steps[72]. The absorption configurations and calculated free energy profiles of OER steps are presented (Figs. 8b-8d). Obviously, the potential rate-determining step (PDS) of NiMoS heterostructures is the third electrochemical step from *O to *OOH with an energy barrier of 1.80 eV. The *OOH species on $NiMoO_x/NiMoS$

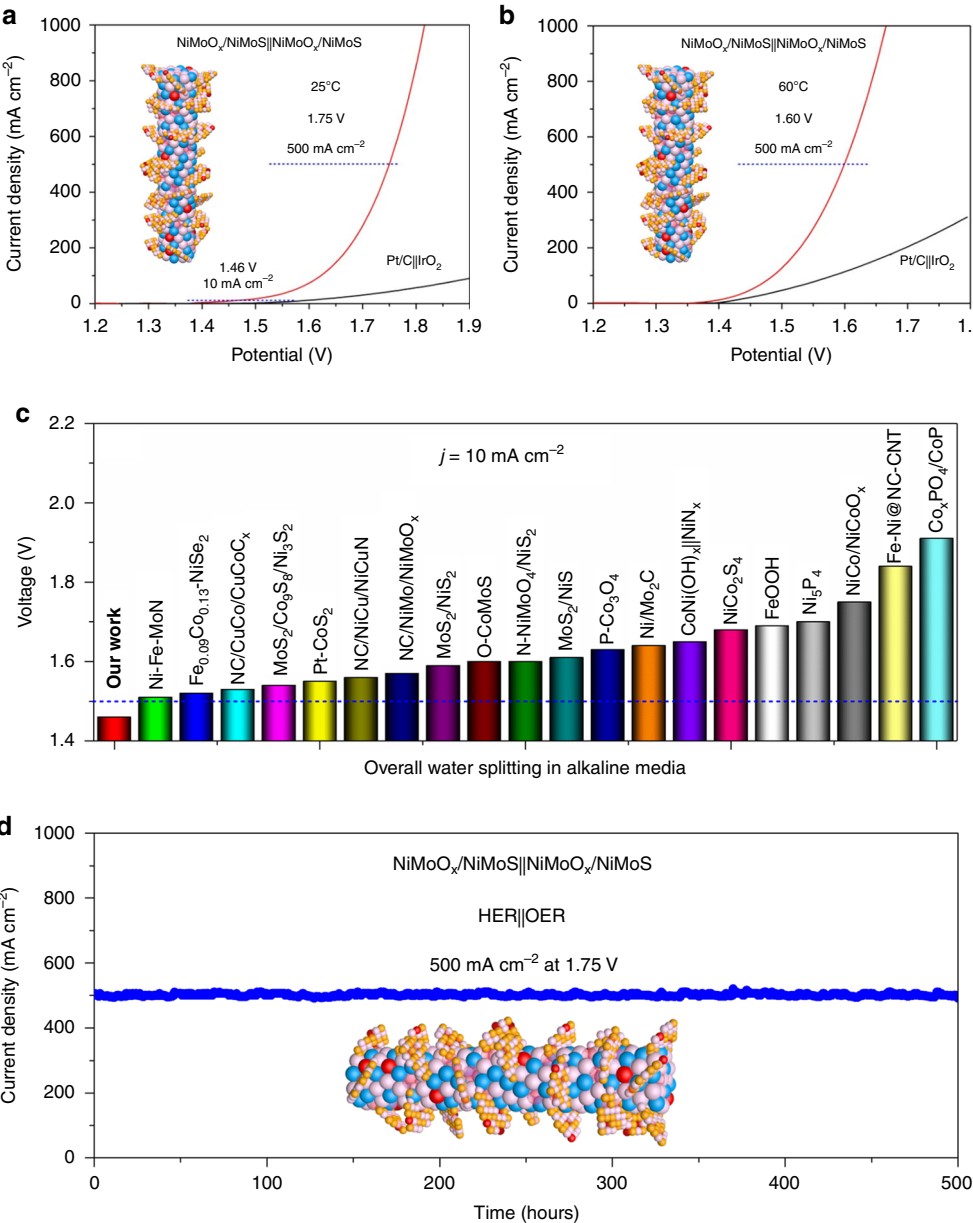

**Fig. 7 Electrocatalytic performance for overall water splitting. a**, **b** Polarization curves by two-electrode system in **a** 1 M KOH at 25 °C and **b** 6 M KOH at 60 °C. **c** Comparison of the cell voltage at 10 mA cm$^{-2}$ for NiMoO$_x$/NiMoS with previously reported catalysts[8,15,20,25,30,42,47,52-64]. **d** Chronoamperometric test at 1.75 V in 1.0 M KOH at 25 °C.

heterostructures are greatly stabilized and overpotential is largely reduced to 0.85 V with the PDS of forming molecule O$_2$. In comparison of NiMoO$_x$/NiMoS, it is of noted that the oxide species in NiO$_x$/Ni$_3$S$_2$ and MoO$_x$/MoS$_2$ heterointerfaces have small evident impact on the decrement of overpotential (Supplementary Fig. 29–32). Therefore, the multi-interfaces of bimetal oxides/sulfides heterostructures are indispensable for the favorable stabilization of intermediates and accelerated electrochemical kinetics. In order to undestand the charge transfer between the NiMoO$_x$/NiMoS interface, charge density difference was performed (Supplementary Fig. 33). It is clear that a remarkable charge transfer across the interface, facilitates the fast electron transfer during the electrocatalytic process. Overall, the theory simulations and experiments demonstrate that the excellent OER and HER activities are facilitated by the synergetic effect of the oxidation/hydrogenation-induced surface reconfiguration.

In this case, the robust electrocatalytic activity is firstly ascribed to 3D hierarchical heterostructures of NiMoO$_x$/NiMoS array owe to the excellent mass transport and gas permeability. Secondly, the constructed interfaces among various heterostructures not only produce together the activities of different materials, but also facilitate the charge transfer and brings exceptionally synergistic effect of typical catalysts by oxidation/hydrogenation-induced surface reconfiguration strategy. Thirdly, the generation of defective species in hierarchical heterostructures could optimize electric conductivity and generate abundant active sites, confirming by low resistances and large ECSAs. Finally, the synergistic effect of the morphology and heterostructure engineering in NiMoO$_x$/NiMoS array promotes the generation of abundant active sites by engineering active sites, optimizing adsorption energies, and accelerating water splitting kinetics. All advantages promote the robust catalytic performance of NiMoO$_x$/NiMoS

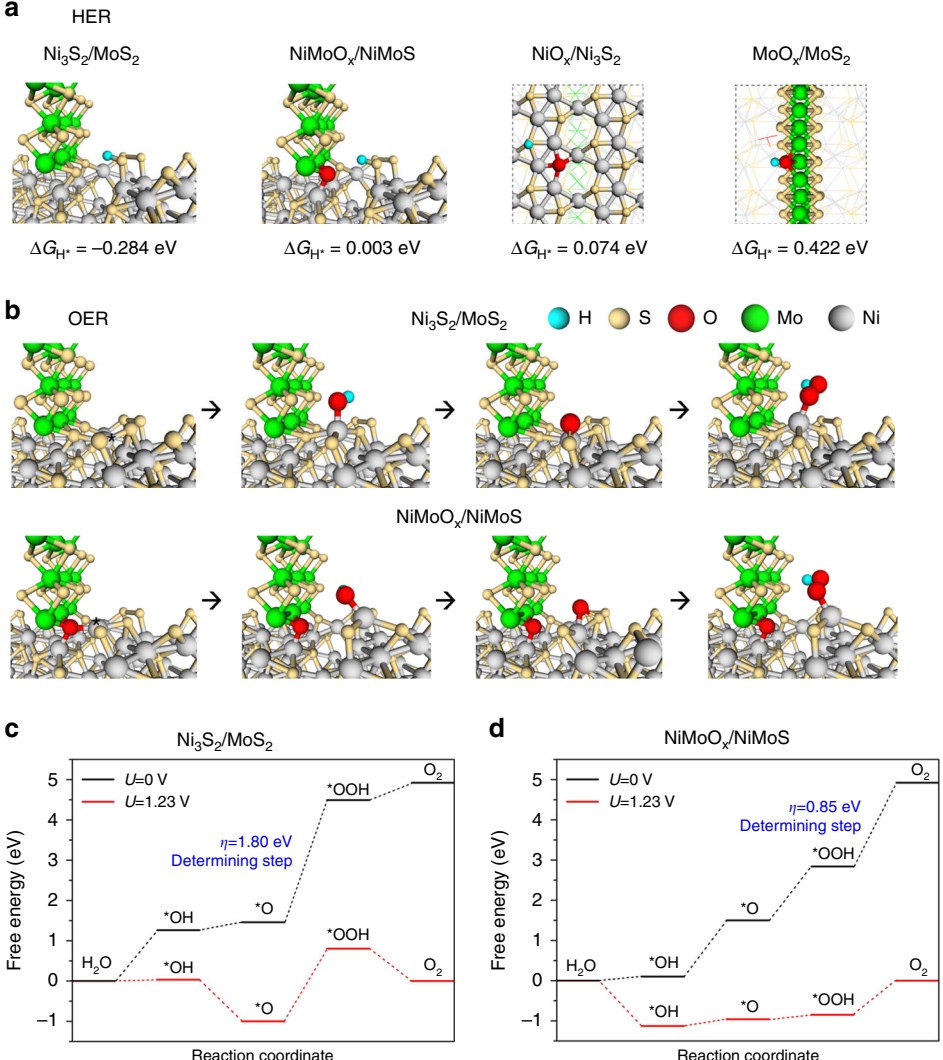

**Fig. 8 Origin of HER/OER activities on NiMoO$_x$/NiMoS. a** Chemisorption models and corresponding Gibbs free energy of H on the interface of Ni$_3$S$_2$/MoS$_2$ and NiMoO$_x$/NiMoS, on the surface of NiO$_x$/Ni$_3$S$_2$ (S) and on the edge of MoO$_x$/MoS$_2$ (Mo). **b** OH, O, and OOH intermediates adsorption configurations for OER on the interface of (top) Ni$_3$S$_2$/MoS$_2$ and (bottom) NiMoO$_x$/NiMoS. **c**, **d** The free energy diagrams for OER on the interface of **c** Ni$_3$S$_2$/MoS$_2$ and (**d**) NiMoO$_x$/NiMoS heterostructures. Cyan, yellow, red, green, and gray balls, respectively, represents H, S, O, Mo, and Ni atoms.

array as a typical catalyst, offering a prospective solution of hierarchical electrocatalysts for practical water splitting applications.

## Discussion

In summary, hierarchical transition bimetal oxides/sulfides array was fabricated by oxidation/hydrogenation-induced surface reconfiguration strategy by use of NiMoS architectures as the precursor, interacting two-dimensional MoO$_x$/MoS$_2$ nanosheets attached to one-dimensional NiO$_x$/Ni$_3$S$_2$ nanorods array. To optimize the electrocatalytic performance, the influence of oxygen plasma power and hydrogenation temperature upon HER and OER performance of NiMoO$_x$/NiMoS array was explored, confirming the best plasma power of 100 W and appropriate thermal treatment temperature at 400 °C. Benefiting from heterostructure engineering, as-synthesized NiMoO$_x$/NiMoS array presents the remarkable electrocatalytic performance, achieving low overpotentials of 38, 89, 174, and 236 mV for HER and 186, 225, 278, and 334 mV for OER at 10, 100, 500 and 1000 mA cm$^{-2}$, even surviving at large current density of 100 and 500 mA cm$^{-2}$ with long-term stability. The extraordinarily enhanced electrocatalytic

performance of transition bimetal oxides/sulfides heterostructure array as the typical model is ascribed to not only the simultaneous modulation of component and geometric structure, but also the systematic optimization of charge transfer, abundant electrocatalytic active sites and exceptionally synergistic effect of heterostructure interfaces. Density functional theory calculations reveal that the coupling interface between NiMoO$_x$ and NiMoS optimizes adsorption energies and accelerates water splitting kinetics, thus promoting the catalytic performance. Especially, the assembled two-electrode cell by use of NiMoO$_x$/NiMoS array delivers the industrially required current densities of 500 and 1000 mA cm$^{-2}$ at record low cell voltages of 1.60 and 1.66 V, along with excellent durability, outperforming most of transition metal-based bifunctional electrocatalysts reported to date. Given hierarchical transition heterostructures array as typical model, this work could open up the avenues to the development of excellent electrocatalysts by engineering active sites for large-scale energy conversion applications.

## Methods

**Materials**. Ni foam was purchased from Suzhou Jiashide Metal Foam Co. Ltd. Ni (NO$_3$)$_2$·6H$_2$O, (NH$_4$)Mo$_7$O$_{24}$·4H$_2$O, thiourea and KOH was purchased from

Aladdin. Pt/C (20 wt% Pt on Vulcan XC-72R) and Nafion (5 wt%) were purchased from Sigma-Aldrich. All chemicals were used as received without further purification. The water used throughout all experiments was purified through a Millipore system.

**Fabrication of NiMoOx/NiMoS heterostructure array**. 0.07 M Ni(NO₃)₂·6H₂O, 0.01 M (NH₄)Mo₇O₂₄·4H₂O and 0.30 M thiourea were dissolved into 15 mL deionized water and stirred for 10 min under room temperature. Then the solution was transferred to a 25 mL Teflon-lined steel autoclave with nickel foam. After hydrothermal reaction at 200 °C for 24 h, NiMoS precursor was obtained through washing with deionized water and then dried in an oven at 60 °C. As-obtained NiMoS precursors were irradiated by RF plasma under an oxygen flow (RF power, 50~150 W) for the oxidation treatment. Afterward the arrays were annealed up to 300–500 °C in H₂/Ar (0.05/0.95) for the typical hydrogenation regulation, thus resulting into the synthesis of NiMoOx/NiMoS heterostructure array by oxidation/ hydrogenation-induced surface reconfiguration strategy. In comparison, as-obtained NiOx/Ni₃S₂ and MoOx/MoS₂ heterostructure arrays were synthesized in parallel by the same procedure as that of NiMoOx/NiMoS array expect for in absence of (NH₄)Mo₇O₂₄·4H₂O or Ni(NO₃)₂·6H₂O in hydrothermal reaction.

**Structural characterization**. Powder XRD patterns of the products were tested with X-ray diffractometer (Japan Rigaku Rotaflex) by Cu K$_\alpha$ radiation ($\lambda = 1.5418$ nm, 40 kV, 40 mA) at room temperature. SEM images of the products were captured by a field-emission scanning electron microscope (SEM, FEI Nova Nano SEM 450). TEM images of the products were performed on transmission electron microscopy (TEM, FEI TF30). The chemical states of the samples were determined by XPS in a Thermo VG ESCALAB250 surface analysis system. The shift of binding energy due to relative surface charging was corrected using the C 1 s level at 284.6 eV as an internal standard.

**Electrochemical measurements**. The electrocatalytic HER and OER performance of different electrocatalysts (1 cm²) were evaluated using a typical three-electrode system in N₂ and O₂-saturated 1 M KOH electrolyte, respectively. All polarization curves at 1 mV s⁻¹ were corrected with iR compensation. The mass loading of NiMoS-based electrocatalysts was tested according to the mass difference. Commercial IrO₂ or 20 wt% Pt/C was dispersed in ethanol solution with Nafion and then the ink was dropped by a micropipettor on Ni foam. The EIS tests were measured by AC impedance spectroscopy at the frequency ranges 10⁶ to 0.1 Hz. According to the Nernst equation ($E_{RHE} = E_{Hg/HgO} + 0.059$ pH $+ 0.098$), where $E_{RHE}$ was the potential vs. a reversible hydrogen potential, $E_{Hg/HgO}$ was the potential vs. Hg/HgO electrode, and pH was the pH value of electrolyte. To determination of Faradaic efficiency, the Faradaic efficiency of HER or OER catalyst is defined as the ratio of the amount of experimentally determined hydrogen or oxygen to that of the theoretically expected hydrogen or oxygen from the HER or OER reaction in 1 M KOH aqueous solution by use of an online gas chromatography system (GC, Techcomp GC 7890 T, Ar carrier gas, Thermo Conductivity Detector). As for the theoretical value, we assumed that 100% current efficiency during the reaction, which means only the HER or OER process was occurring at the working electrode. The theoretically expected amount of hydrogen or oxygen was then calculated by applying the Faraday law, which states that the passage of 96485.4 C causes 1 equivalent of reaction.

**First-principle calculations**. Density functional theory calculations were carried out by the Vienna ab initio simulation package (VASP), using the planewave basis with an energy cutoff of 400 eV, the projector augmented wave pseudopotentials, and the generalized gradient approximation parameterized by Perdew, Burke, and Ernzerhof (GGA-PBE) for exchange-correlation functional[73]. The Brillouin zones of the supercells were sampled by $4 \times 4 \times 1$ uniform k point mesh. With fixed cell parameters, the model structures were fully optimized using the convergence criteria of $10^{-5}$ eV for the electronic energy and $10^{-2}$ eV/Å for the forces on each atom. The supercells dimension in x and y was 11.598 Å and 12.243 Å, respectively. The vacuum region in the z direction was adopted large than 15 Å so that the spurious interactions of neighboring models are negligible. Then O atom was used to replace the S atom on the edge of MoS₂ and the surface of Ni₃S₂ and the interface of MoS₂ and Ni₃S₂, respectively[74]. To simulate the edge, the surface and interface incorporate with the oxides. Both spin-polarized and spin-unpolarized computations were performed. The computational results show that both NiMoS and NiMoOx/NiMoS are magnetic. In addition, we applied the DFT-D3 (BJ) method to evaluate the van der Waals (vdW) effect in all calculations.

The Gibbs free energy of the intermediates for HER and OER process, that is, H, OH, O, and OOH, can be calculated as[75,76]

$$\Delta G = E_{ads} + \Delta E_{ZPE} - T\Delta S \qquad (1)$$

where $E_{ads}$ is the adsorption energy of intermediate, $\Delta E_{ZPE}$ is the zero point energy difference between the adsorption state and gas state, $T$ is the temperature, and $\Delta S$ is the entropy various between the adsorption and gas phase. For adsorbates, $E_{ZPE}$ and $S$ are obtained from vibrational frequencies calculations with harmonic approximation and contributions from the slabs are neglected, whereas for

molecules these values are taken from NIST-JANAF thermochemical Tables[77]. The contributions are listed (Supplementary Table 8). Usually, the vibration entropy of hydrogen adsorption on the substrate is small, the entropy of hydrogen adsorption is $\Delta S \approx -1/2 S^\circ$, where $S^0$ is the entropy of H₂ in the gas phase at the standard conditions. The corrected for free energy equation was defined by

$$\Delta G = E_{ads} + 0.24\ eV \qquad (2)$$

The intermediates adsorption energy $E_{ads}$ for *H, *OH, *O, and *OOH can be used as DFT ground state energy calculated as

$$\Delta E_{*H} = E(^*H) - E(^*) - 1/2E(H_2) \qquad (3)$$

$$\Delta E_{*OOH} = E(^*OOH) - E(^*) - (2E_{H_2O} - 3/2E_{H_2}) \qquad (4)$$

$$\Delta E_{*O} = E(^*O) - E(^*) - (E_{H_2O} - E_{H_2}) \qquad (5)$$

$$\Delta E_{*OH} = E(^*OH) - E(^*) - (E_{H_2O} - 1/2E_{H_2}) \qquad (6)$$

The OER process in alkaline medium generally occur through the following steps:

$$* + OH^- \rightarrow OH^* + e^- \qquad \Delta G_1 \qquad (7)$$

$$OH^* + OH^- \rightarrow\ ^*O + H_2O + e^- \qquad \Delta G_2 \qquad (8)$$

$$^*O + OH^- \rightarrow OOH^* + e^- \qquad \Delta G_3 \qquad (9)$$

$$OOH^* + OH^- \rightarrow O_2 + H_2O + e^- + * \qquad \Delta G_4 \qquad (10)$$

where * denotes adsorption active site on the substrate.

$$\Delta G_1 = \Delta G_{*OH} \qquad (11)$$

$$\Delta G_2 = \Delta G_{*O} - \Delta G_{*OH} \qquad (12)$$

$$\Delta G_3 = \Delta G_{*OOH} - \Delta G_{*O} \qquad (13)$$

$$\Delta G_4 = 4.92 - \Delta G_{*OOH} \qquad (14)$$

The overpotential η is defined as

$$\eta = \max\{\Delta G_1, \Delta G_2, \Delta G_3, \Delta G_4\} - 1.23\ eV \qquad (15)$$

## Data availability
The data that support the findings of this work are available from the corresponding author upon reasonable request.

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

## Acknowledgements

This work was supported by National Natural Science Foundation of China (Nos. 21972015, 51672034), Young top talents project of Liaoning Province (No. XLYC1907147), Joint Research Fund Liaoning-Shenyang National Laboratory for Materials Science (No. 2019JH3/30100003), the Fundamental Research Funds for the Central Universities (No. DUT20TD06), the Swedish Research Council, and the K&A Wallenberg Foundation.

## Author contributions

J.H. supervised this study. J.H., P.Z., and Y.W. conceived the idea. P.Z. and Y.W. planned and carried out the experiments, collected, and analyzed the experimental data. S.C. performed SEM and TEM characterizations. Y.Z. and J.G. conducted theoretical calculations. P.Z., Y.W., and J.H. wrote the paper. All the authors have discussed the results and wrote the paper together.

## Competing interests

The authors declare no competing interests.
