## [Peer Review File · Nature Communications]

REVIEWER COMMENTS

Reviewer #1 (Remarks to the Author):

The authors reported the experimental and computational results on electrocatalytic activities of hierarchical defective transition bimetal oxides/sulfides heterostructure array as the promising electrocatalyst for hydrogen evolution reaction (HER) and oxygen evolution reaction (OER) as well as overall water splitting in alkaline media.

I recommend it to publish in Nature Communication after some revisions, detailed suggestions are listed as following.

Detailed comments:

[1] In the experimental section, the author should provide the details for electrochemical measurements of HER and OER, is there pretreatment in H₂ or O₂ saturated alkaline condition before HER and OER tests?

[2] The novelty of this work must be further emphasized, highlighting the roles of engineering interface atomic activation sites on hierarchical defective transition bimetal oxides/sulfides heterostructure array. In addition, the "outstanding electrocatalytic activity" of the bimetal oxides/sulfides heterostructure array is the least important aspect of the paper, and should be put into better context by comparing to known reported systems.

[3] Although XPS spectra has been well conducted for NiMoOx/NiMoS, could the author also give the detailed characterization of MoOx/MoS₂ and NiOx/Ni₃S₂ by XPS spectra?

[4] Those measured current densities were all calculated with respect to the geometric areas of the assembled electrodes, not directly related to the intrinsic activities of these catalyst samples. In order to show whether the intrinsic activity is improved or not, turnover frequency (TOF) value of both HER and OER should be calculated and compared with those of other catalysts.

[5] In the manuscript, it is very impressive that the very large catalytic current densities for both HER and OER are driven by the low potentials. In this case, the Faraday efficiencies of OER, HER and overall water splitting should be given, confirming the real electrocatalytic activities of NiMoOx/NiMoS array.

[6] The intrinsic activity should play an important role in electrocatalysis, so it is more valuable to take this into account, in-depth understanding the real electrocatalytic process in this work.

[7] There are some reports about the electrocatalysts, delivering large catalytic current densities (e.g., 500 and 1000 mA cm⁻²) for practical application. Can the authors make a comparison among the catalytic activities of this catalyst in the manuscript and these reported catalysts?

[8] The 'Discussion' section looks like 'Conclusions' in this work. The authors need to comprehensively discuss on their study and new findings if the results and discussion can be split in this manuscript.

Reviewer #2 (Remarks to the Author):

In this manuscript, hierarchical transition bimetal oxides/sulfides arrays (NiMoOx/NiMoS) were fabricated by an oxidation/hydrogenation-induced surface reconfiguration strategy and investigated for water splitting. A suite of electrochemical analysis and characterizations clearly indicate the formation of 3D, defect-rich heterostructure. Impressive electrocatalytic performance for HER, OER and overall water splitting were demonstrated by thorough electrochemical tests. Active sites and reaction mechanism were further revealed via DFT studies. This work provides an attractive strategy of morphology, defect and heterostructure engineering upon hierarchical oxide/sulfides for water splitting, and is thus publishable. However, there are also several questions to be addressed.

1. In introduction section, the shortcomings and remaining problems of reported catalysts are suggested to highlight. In addition to enumerate relevant materials, the basis and reason for the

design and selection of the catalyst and the structure should be explained more clearly.

2. Some results of the experimental characterization need to be checked. For example:

(1) For the results of TEM analysis on defect structures marked in Figure 3e, the size of each white dot looks much bigger than the size of an atom. Are these dots (supposed to be defects) attributed to electron beam irradiation damage or derived from the material/sample itself?

(2) Please check the lattice fringes for Ni₃S₂ marked in Figure 3e. The fringes are horizontal to the edge but interlayer distance is marked vertical.

(3) The images in figure S5 are identical to that in Figure 3f-i. It's not necessary to repeat this information.

3. In the part of theoretical calculation, the additional explanations for the established catalyst models and the exploration of atomic activation sites at the interface need to be supplemented. (1) The modeling of metal oxides is described as "used O atom to replace the S atom on the edge of MoS₂ and the surface of Ni₃S₂ and the interface of MoS₂ and Ni₃S₂, respectively. To simulate the edge, surface and interface incorporate with oxides how to effects on the catalyst overall water splitting". The crystal facets of MoO₂ and MoO₃ can be observed from TEM, and the diffraction peaks of MoO₂, MoO₃ and NiO can be obtained from XRD results. The crystal structure information of metal oxides can be confirmed by the above results, but the theoretical models of NiO_x, MoO_x and NiMoO_x are represented by an O atom instead of S atom in Ni₃S₂, MoS₂ and NiMoS, respectively, which might not embody "multi-interfaces of dual-metal based oxide-sulfide heterostructures" claimed in the article. Please rethink the deviations between the theoretical models and structures characterized in experiments.

(2) The manuscript proposed that the constructed interfaces among various heterostructures facilitate the charge transfer and bring exceptionally synergistic effect by oxidation/hydrogenation-induced surface reconfiguration. However, the charge transfer of NiMoO_x/NiMoS in the catalytic process has not been given from the theoretical calculation point of view, so it would be better to make a further explanation.

(3) The claim "the synergistic effect of the morphology, defect and heterostructure engineering in NiMoO_x/NiMoS array promote the generation of abundant active sites by engineering interface atomic activation sites, optimizing adsorption energies and accelerating water splitting kinetics" has been emphasized in the manuscript. However, the theoretical calculation does not reflect the characteristics of the defective structure, nor does it describe the influence of the defective structure on its catalytic performance.

(4) The O atoms in NiO_x, MoO_x, NiMoO_x are possible sites to involve in OER. If so, they may interact and is competitive with other active site, which would change the favored potential energy surface. Discussion along this line is suggested.

Reviewer #3 (Remarks to the Author):

Authors presented a non-PGM catalyst for water splitting. I think the activities reported herein are quite high, but I have some major concerns. I suggest major revision and authors should address the following issues before the articles publication. Please see the comments below:

1. I am not convinced about the measurements of surface area; I suggest that authors should elaborate on it. What were the values for measured electroactive surface area, what were the geometric areas? Do they correspond reasonably with each other, considering the porosity of nickel foams?

2. Since the determination of ECSA in metal oxide catalysts is not precise, I suggest the authors report also the activities observed per mass, and compare the activities per mass with the literature mass activities.

3. Authors should report how many times the activities were reproduced and should include the errors bars to ensure the reproducibility of the results

4. The Figure 7 shows the polarization curve for OER not the overall water splitting, please correct. Also please report where were the values for figure 7c were taken from, were they done in exact same conditions (electrolyte etc.).

5. I am wondering if they had any difficulties with bubble formation and blocking of the catalyst active site.
6. On Figure 6d, what is meant by 100mA/cm² at 225mV, does that potential correspond to the potential vs RHE when 100mA/cm² was observed? I have the same question for potential of 500mA/cm² at 278mV.
7. Figure 1 please label which colored ball corresponds to which element.
8. Figure 2 for the elemental mapping images, which image corresponds to which element please label.
9. In overall the written English of the article needs major improvements. There are many minor mistakes such as on Page 15: NiMoOx/NiMoS array was utilized as as cathode and anode in two-electrode...

Editorial notes:

Reviewer comments & decisions:

REVIEWER COMMENTS

Reviewer #1 (Remarks to the Author):

The authors reported the experimental and computational results on electrocatalytic activities of hierarchical defective transition bimetal oxides/sulfides heterostructure array as the promising electrocatalyst for hydrogen evolution reaction (HER) and oxygen evolution reaction (OER) as well as overall water splitting in alkaline media. I recommend it to publish in Nature Communication after some revisions, detailed suggestions are listed as following.

Detailed comments:

[1] In the experimental section, the author should provide the details for electrochemical measurements of HER and OER, is there pretreatment in H₂ or O₂ saturated alkaline condition before HER and OER tests?

[2] The novelty of this work must be further emphasized, highlighting the roles of engineering interface atomic activation sites on hierarchical defective transition bimetal oxides/sulfides heterostructure array. In addition, the “outstanding electrocatalytic activity” of the bimetal oxides/sulfides heterostructure array is the least important aspect of the paper, and should be put into better context by comparing to known reported systems.

[3] Although XPS spectra has been well conducted for NiMoO_x/NiMoS, could the author also give the detailed characterization of MoO_x/MoS₂ and NiO_x/Ni₃S₂ by XPS spectra?

[4] Those measured current densities were all calculated with respect to the geometric areas of the assembled electrodes, not directly related to the intrinsic activities of these catalyst samples. In order to show whether the intrinsic activity is improved or not, turnover frequency (TOF) value of both HER and OER should be calculated and compared with those of other catalysts.

[5] In the manuscript, it is very impressive that the very large catalytic current densities for both HER and OER are driven by the low potentials. In this case, the Faraday efficiencies of OER, HER and overall water splitting should be given, confirming the real electrocatalytic activities of NiMoO_x/NiMoS array.

[6] The intrinsic activity should play an important role in electrocatalysis, so it is more valuable to take this into account, in-depth understanding the real electrocatalytic process in this work.

[7] There are some reports about the electrocatalysts, delivering large catalytic current densities (e.g., 500 and 1000 mA cm⁻²) for practical application. Can the authors make a comparison among the catalytic activities of this catalyst in the manuscript and these reported catalysts?

[8] The 'Discussion' section looks like 'Conclusions' in this work. The authors need to comprehensively discuss on their study and new findings if the results and discussion can be split in this manuscript.

Reviewer #2 (Remarks to the Author):

In this manuscript, hierarchical transition bimetal oxides/sulfides arrays (NiMoO_x/NiMoS) were fabricated by an oxidation/hydrogenation-induced surface reconfiguration strategy and investigated for water splitting. A suite of electrochemical analysis and characterizations clearly indicate the formation of 3D, defect-rich heterostructure. Impressive electrocatalytic performance for HER, OER and overall water splitting were demonstrated by thorough electrochemical tests. Active sites and reaction mechanism were further revealed via DFT studies. This work provides an attractive strategy of morphology, defect and heterostructure engineering upon hierarchical oxide/sulfides for water splitting, and is thus publishable. However, there are also several questions to be addressed.

[1] In introduction section, the shortcomings and remaining problems of reported catalysts are suggested to highlight. In addition to enumerate relevant materials, the basis and reason for the design and selection of the catalyst and the structure should be explained more clearly.

[2] Some results of the experimental characterization need to be checked. For example:

(1) For the results of TEM analysis on defect structures marked in Figure 3e, the size of each white dot looks much bigger than the size of an atom. Are these dots (supposed to be defects) attributed to electron beam irradiation damage or derived from the material/sample itself?

(2) Please check the lattice fringes for Ni_3S_2 marked in Figure 3e. The fringes are horizontal to the edge but interlayer distance is marked vertical.

(3) The images in figure S5 are identical to that in Figure 3f-i. It's not necessary to repeat this information.

[3] In the part of theoretical calculation, the additional explanations for the established catalyst models and the exploration of atomic activation sites at the interface need to be supplemented.

(1) The modeling of metal oxides is described as "used O atom to replace the S atom on the edge of MoS_2 and the surface of Ni_3S_2 and the interface of MoS_2 and Ni_3S_2 , respectively. To simulate the edge, surface and interface incorporate with oxides how to effects on the catalyst overall water splitting". The crystal facets of MoO_2 and MoO_3 can be observed from TEM, and the diffraction peaks of MoO_2 , MoO_3 and NiO can be obtained from XRD results. The crystal structure information of metal oxides can be confirmed by the above results, but the theoretical models of NiO_x , MoO_x and NiMoO_x are represented by an O atom instead of S atom in Ni_3S_2 , MoS_2 and NiMoS , respectively, which might not embody "multi-interfaces of dual-metal based oxide-sulfide heterostructures" claimed in the article. Please rethink the deviations between the theoretical models and structures characterized in experiments.

(2) The manuscript proposed that the constructed interfaces among various heterostructures facilitate the charge transfer and bring exceptionally synergistic effect by oxidation/hydrogenation-induced surface reconfiguration. However, the charge transfer of $\text{NiMoO}_x/\text{NiMoS}$ in the catalytic process has not been given from the theoretical calculation point of view, so it would be better to make a further explanation.

(3) The claim "the synergistic effect of the morphology, defect and heterostructure engineering in $\text{NiMoO}_x/\text{NiMoS}$ array promote the generation of abundant active sites by engineering interface atomic activation sites, optimizing adsorption energies and accelerating water splitting kinetics" has been emphasized in the manuscript. However, the theoretical calculation does not reflect the characteristics of the defective structure, nor does it describe the influence of the defective structure on its catalytic performance.

(4) The O atoms in NiO_x , MoO_x , NiMoO_x are possible sites to involve in OER. If so, they

may interact and is competitive with other active site, which would change the favored potential energy surface. Discussion along this line is suggested.

Reviewer #3 (Remarks to the Author):

Authors presented a non-PGM catalyst for water splitting. I think the activities reported herein are quite high, but I have some major concerns. I suggest major revision and authors should address the following issues before the articles publication.

Please see the comments below:

[1] I am not convinced about the measurements of surface area; I suggest that authors should elaborate on it. What were the values for measured electroactive surface area, what were the geometric areas? Do they correspond reasonably with each other, considering the porosity of nickel foams?

[2] Since the determination of ECSA in metal oxide catalysts is not precise, I suggest the authors report also the activities observed per mass, and compare the activities per mass with the literature mass activities.

[3] Authors should report how many times the activities were reproduced and should include the errors bars to ensure the reproducibility of the results.

[4] The Figure 7 shows the polarization curve for OER not the overall water splitting, please correct. Also please report where were the values for figure 7c were taken from, were they done in exact same conditions (electrolyte etc.).

[5] I am wondering if they had any difficulties with bubble formation and blocking of the catalyst active site.

[6] On Figure 6d, what is meant by $100\text{mA}/\text{cm}^2$ at 225mV , does that potential correspond to the potential vs RHE when $100\text{mA}/\text{cm}^2$ was observed? I have the same question for potential of $500\text{mA}/\text{cm}^2$ at 278mV .

[7] Figure 1 please label which colored ball corresponds to which element.

[8] Figure 2 for the elemental mapping images, which image corresponds to which

element please label.

[9] In overall the written English of the article needs major improvements. There are many minor mistakes such as on Page 15: NiMoO_x/NiMoS array was utilized as as cathode and anode in two-electrode...

Responses to the reviewer's comments

Reviewer #1 (Remarks to the Author):

The authors reported the experimental and computational results on electrocatalytic activities of hierarchical defective transition bimetal oxides/sulfides heterostructure array as the promising electrocatalyst for hydrogen evolution reaction (HER) and oxygen evolution reaction (OER) as well as overall water splitting in alkaline media. I recommend it to publish in Nature Communication after some revisions, detailed suggestions are listed as following.

We are grateful to the reviewer's comments. We have made careful revisions according to each comment, as summarized below.

Detailed comments:

[1] In the experimental section, the author should provide the details for electrochemical measurements of HER and OER, is there pretreatment in H₂ or O₂ saturated alkaline condition before HER and OER tests?

Response : Thanks for the comments and suggestions from the reviewer. In the experimental section, the electrocatalytic HER and OER performance of different electrocatalysts were evaluated using a typical three-electrode system in N₂ and O₂-saturated 1 M KOH electrolyte, respectively. We have added these contents to the revised manuscript in the experimental section.

[2] The novelty of this work must be further emphasized, highlighting the roles of engineering interface atomic activation sites on hierarchical defective transition bimetal oxides/sulfides heterostructure array. In addition, the “outstanding electrocatalytic activity” of the bimetal oxides/sulfides heterostructure array is the least important aspect of the paper, and should be put into better context by comparing to known reported systems.

Response : Thanks for the comments and suggestions from the reviewer. The novelty of this work has been emphasized, highlighting the roles of engineering interface atomic activation sites on defective transition bimetal oxides/sulfides heterostructure array. In this case, the robust electrocatalytic activity is firstly ascribed to three-dimensional hierarchical heterojunctions of NiMoO_x/NiMoS array supported on 3D Ni foam owe to the

Table R1. Comparison of overall water splitting activities of bifunctional electrocatalysts.

Catalysts	Electrolyte	Potential at 10 mA cm ⁻²	References
MoS ₂ /NiS ₂ nanosheets	1 M KOH	1.59 V	[1]
O-CoMoS	1 M KOH	1.60 V	[2]
MoS ₂ /NiS NCs	1 M KOH	1.61 V	[3]
MoS ₂ /Co ₉ S ₈ /Ni ₃ S ₂ /Ni	1 M KOH	1.54 V	[4]
δ-FeOOH NSs/NF	1 M KOH	1.69 V	[5]
P-Co ₃ O ₄ /NF	1 M KOH	1.63 V	[6]
Ni/Mo ₂ C nanoparticles	1 M KOH	1.64 V	[7]
Pt-CoS ₂ /CC	1 M KOH	1.55 V	[8]
Fe-Ni@NC-CNTs	1 M KOH	1.84V	[9]
NC/CuCo/CuCoO _x	1 M KOH	1.53 V	[10]
Ni _x Co _{3-x} O ₄ @NiCo/NiCoO _x	1 M KOH	1.75 V	[11]
NC/NiCu/NiCuN	1 M KOH	1.56V	[12]
NiCo ₂ S ₄	1 M KOH	1.68 V	[13]
NC/NiMo/NiMoO _x	1 M KOH	1.57 V	[14]
Ni ₅ P ₄	1 M KOH	1.70 V	[15]
FeP	1 M KOH	1.69 V	[16]
CoNi(OH) _x NiN _x	1 M KOH	1.65 V	[17]
Co _x PO ₄ /CoP	1 M KOH	1.91 V	[18]
Fe-Ni-MoN	1 M KOH	1.51 V	[19]
Fe _{0.09} Co _{0.13} -NiSe ₂	1 M KOH	1.52 V	[20]
N-NiMoO ₄ /NiS ₂	1 M KOH	1.6 V	[21]
NiMoO _x /NiMoS	1 M KOH	1.46 V	This work

excellent mass transport and gas permeability. Secondly, the constructed interfaces among various heterostructures not only produce together the activities of different materials, but also facilitate the charge transfer and brings exceptionally synergistic effect of typical catalysts by novel oxidation/hydrogenation-induced surface reconfiguration strategy. Thirdly, the generation of defective oxides and sulfides species in hierarchical heterostructures with a large number of the defects could optimize electric conductivity and generate abundant active sites, confirming by low resistances and large ECSAs. Finally, the synergistic effect of the morphology, defect and heterostructure engineering in NiMoO_x/NiMoS array promote the generation of abundant active sites by engineering interface atomic activation sites, optimizing adsorption energies and accelerating water splitting kinetics. All advantages promote the robust catalytic performance of defective NiMoO_x/NiMoS array as a typical catalyst, offering a prospective solution of hierarchical electrocatalysts for practical water splitting application.

Figure R1. Comparison of the cell voltages at 10 mA cm⁻² for NiMoO_x/NiMoS heterostructure array as the typical catalysts with previously reported electrocatalysts.

With regard to the statement of “outstanding electrocatalytic activity” of the bimetal oxides/sulfides heterostructure array as the least important aspect of the paper, the relative statement has been put into the revised manuscript. In comparison of different electrocatalysts, such as Ni-Fe-MoN,¹⁹ Fe_{0.09}Co_{0.13}-NiSe₂,²⁰ NC/CoCu/CoCuO_x,¹⁰ MoS₂/Co₉S₈/Ni₃S₂,⁴ O-CoMoS,² N-NiMoO₄/NiS₂,²¹ MoS₂/NiS,³ P-Co₃O₄,⁶ Ni/Mo₂C,⁷ CoNi(OH)_x/NiN_x,¹⁷ NiCo₂S₄,¹³ FeOOH,⁵ Ni₅P₄,¹⁵ NiCo/NiCoO_x,¹¹ Fe-Ni@NC-CNT,⁹ Co_xPO₄/CoP¹⁸ and commercial Pt/C||IrO₂ electrodes (Figure R1), the lower voltage at 10 mA cm⁻² was obtained by NiMoO_x/NiMoS array. Typically, the record low voltages of 1.60 and 1.66 V of two-electrode system in 6 M KOH solution at 60 °C were achieved for the industrial current densities of 500 and 1000 mA cm⁻², respectively, and it is still better

than that of Pt/C||IrO₂ couple. With regard to the operating stability as important metric, this typical two-electrode cell can maintain the excellent electrocatalytic activity at a large current density of 500 mA cm⁻² over 500 h without obvious degradation in 1 M KOH solution at 25 °C.

References:

- (1) Lin, J. *et al.* Defect-Rich Heterogeneous MoS₂/NiS₂ Nanosheets Electrocatalysts for Efficient Overall Water Splitting. *Adv. Sci.* **6**, 1900246 (2019).
- (2) Hou, J. *et al.* Vertically Aligned Oxygenated-CoS₂-MoS₂ Heteronanosheet Architecture from Polyoxometalate for Efficient and Stable Overall Water Splitting. *ACS Catal.* **8**, 4612-4621 (2018).
- (3) Zhai, Z. *et al.* Dimensional Construction and Morphological Tuning of Heterogeneous MoS₂/NiS Electrocatalysts for Efficient Overall Water Splitting. *J. Mater. Chem. A* **6**, 9833-9838 (2018).
- (4) Yang, Y. *et al.* Hierarchical Nanoassembly of MoS₂/Co₉S₈/Ni₃S₂/Ni as a Highly Efficient Electrocatalyst for Overall Water Splitting in a Wide pH Range. *J. Am. Chem. Soc.* **141**, 10417-10430 (2019).
- (5) Liu, B. *et al.* Iron Vacancies Induced Bifunctionality in Ultrathin Ferrosulfide Nanosheets for Overall Water Splitting. *Adv. Mater.* **30**, 1803144 (2018).
- (6) Wang, Z. *et al.* Phosphorus-Doped Co₃O₄ Nanowire Array: A Highly Efficient Bifunctional Electrocatalyst for Overall Water Splitting. *ACS Catal.* **8**, 2236-2241 (2018).
- (7) Li, M. *et al.* Ni Strongly Coupled with Mo₂C Encapsulated in Nitrogen-Doped Carbon Nanofibers as Robust Bifunctional Catalyst for Overall Water Splitting. *Adv. Energy Mater.* **9**, 1803185 (2019).
- (8) Han, X. *et al.* Ultrafine Pt Nanoparticle-Decorated Pyrite-Type CoS₂ Nanosheet Arrays Coated on Carbon Cloth as a Bifunctional Electrode for Overall Water Splitting. *Adv. Energy Mater.* **8**, 1800935 (2018).
- (9) Zhao, X. *et al.* Bifunctional Electrocatalysts for Overall Water Splitting from an Iron/Nickel-Based Bimetallic Metal-Organic Framework/Dicyandiamide Composite. *Angew. Chem. Int. Ed.* **57**, 8921-8926 (2018).
- (10) Hou, J. *et al.* Electrical Behavior and Electron Transfer Modulation of Nickel-Copper Nanoalloys Confined in Nickel-Copper Nitrides Nanowires Array Encapsulated in Nitrogen-Doped Carbon Framework as Robust Bifunctional Electrocatalyst for Overall Water Splitting. *Adv. Funct. Mater.* **28**, 1704447 (2018).
- (11) Yan, X. *et al.* From Water Oxidation to Reduction: Transformation from Ni_(x)Co_(3-x)O₄ Nanowires to NiCo/NiCoO_(x) Heterostructures. *ACS Appl. Mater. Interfaces* **8**, 3208-3214

(2016).

(12) Hou, J., Sun, Y., Wu, Y., Cao, S. & Sun, L. Promoting Active Sites in Core-Shell Nanowire Array as Mott-Schottky Electrocatalysts for Efficient and Stable Overall Water Splitting. *Adv. Funct. Mater.* **28**, 1803278 (2018).

(13) Sivanantham, A., Ganesan, P. & Shanmugam, S. Hierarchical NiCo₂S₄ Nanowire Arrays Supported on Ni Foam: An Efficient and Durable Bifunctional Electrocatalyst for Oxygen and Hydrogen Evolution Reactions. *Adv. Funct. Mater.* **26**, 4661-4672 (2016).

(14) Hou, J., Wu, Y., Cao, S., Sun, Y. & Sun, L. Active Sites Intercalated Ultrathin Carbon Sheath on Nanowire Arrays as Integrated Core-Shell Architecture: Highly Efficient and Durable Electrocatalysts for Overall Water Splitting. *Small* **13**, 1702018 (2017).

(15) Ledendecker, M. *et al.* The Synthesis of Nanostructured Ni₅P₄ Films and Their Use as a Non-noble Bifunctional Electrocatalyst for Full Water Splitting. *Angew. Chem. Int. Ed.* **54**, 12361-12365 (2015).

(16) Yan, Y. *et al.* A Flexible Electrode Based on Iron Phosphide Nanotubes for Overall Water Splitting. *Chem. Eur. J.* **21**, 18062-18067 (2015).

(17) Li, S. *et al.* Co-Ni-Based Nanotubes/Nanosheets as Efficient Water Splitting Electrocatalysts. *Adv. Energy Mater.* **6**, 1501661 (2016).

(18) Yang, Y. *et al.* Porous Cobalt-based Thin Film as a Bifunctional Catalyst for Hydrogen Generation and Oxygen Generation. *Adv. Mater.* **27**, 3175-3180 (2015).

(19) Zhu, C. *et al.* Fe-Ni-Mo Nitride Porous Nanotubes for Full Water Splitting and Zn-Air Batteries. *Adv. Energy Mater.* **8**, 1802327 (2018).

(20) Sun, Y. *et al.* Strong Electronic Interaction in Dual-Cation-Incorporated NiSe₂ Nanosheets with Lattice Distortion for Highly Efficient Overall Water Splitting. *Adv. Mater.* **30**, 1802121 (2018).

(21) An, L. *et al.* Epitaxial Heterogeneous Interfaces on N-NiMoO₄/NiS₂ nanowires/Nanosheets to Boost Hydrogen and Oxygen Production for Overall Water Splitting. *Adv. Funct. Mater.* **29**, 1805298 (2019).

[3] Although XPS spectra have been well conducted for NiMoO_x/NiMoS, could the author also give the detailed characterization of MoO_x/MoS₂ and NiO_x/Ni₃S₂ by XPS spectra?

Response : Thanks for the comments and suggestions from the reviewer. To conduct the chemical valences of the catalysts, X-ray photoelectron spectroscopy (XPS) spectrum has been tested. We have added these contents to the revised manuscript.

Figure R2. XPS spectra of as-prepared $\text{MoO}_x/\text{MoS}_2$ array. (a) Mo 3d, (b) S 2p and (c) O 1s.

With regard to Mo 3d regions, the main peak could be split into two distinctive peaks of Mo $3d_{5/2}$ (229.1 eV) and Mo $3d_{3/2}$ (232.4 eV), indicating the dominance of Mo^{4+} in NiMoS .¹ The peaks at 855.2, 861.5, 872.9 and 879.5 eV can be indexed to Ni $2p_{3/2}$ and Ni $2p_{1/2}$ orbitals as well as two satellites in NiMoS .² However, the signals at 229.3, 232.4 and 235.5 eV can be indexed to Mo^{4+} $3d_{5/2}$, $\text{Mo}^{4+/6+}$ $3d_{3/2}$ and Mo^{6+} $3d_{3/2}$ orbitals, confirming the existence of Mo^{4+} and Mo^{6+} in $\text{NiMoO}_x/\text{NiMoS}$ owe to the formation of MoO_x .³ For Ni 2p orbitals, there is a shift upon the peak positions and the two new peaks at 854.6 and 852.6 eV, demonstrating the existence of Ni–O bonds and metallic Ni^0 and thus indicating the formation of NiO_x species in $\text{NiMoO}_x/\text{NiMoS}$.^{3,4} Typically, the signals at 529.5 and 531.5 eV for O 1s belong to typical metal-oxygen bonds and oxygen vacancies in hierarchical defective $\text{NiMoO}_x/\text{NiMoS}$ heterostructure.⁴ With regard to S 2p peaks, the negative shift was observed in hierarchical defective $\text{NiMoO}_x/\text{NiMoS}$ with the increasing temperature of thermal treatment, demonstrating the loss of S and the formation of sulfur

vacancies.⁵ The similar phenomenon of O 1s and S 2p was observed in MoO_x/MoS₂ and NiO_x/Ni₃S₂ heterostructures. With regard to XPS spectra of MoO_x/MoS₂ array and NiO_x/Ni₃S₂ array (Figure R2 and Figure R3), the signals at 229.3, 232.4 and 235.5 eV can be indexed to Mo⁴⁺ 3d_{5/2}, Mo^{4+/6+} 3d_{3/2} and Mo⁶⁺ 3d_{3/2} orbitals,¹ confirming the existence of Mo⁴⁺ and Mo⁶⁺ in MoO_x/MoS₂ array. While two new peaks at 854.6 and 852.6 eV were observed,^{3,4} demonstrating the existence of Ni–O bonds and metallic Ni⁰ and thus indicating the formation of NiO_x species in NiO_x/Ni₃S₂. Thus, the combined analysis demonstrated the successful synthesis of hierarchical defective NiMoO_x/NiMoS heterostructure array.

Figure R3. XPS spectra of as-prepared NiO_x/Ni₃S₂ array. (a) Ni 3d, (b) S 2p and (c) O 1s.

References:

(1) Hou, J. et al. Vertically Aligned Oxygenated-CoS₂-MoS₂ Heteronanoshet Architecture from Polyoxometalate for Efficient and Stable Overall Water Splitting. *ACS Catal.* **8**, 4612-4621 (2018).

- (2) Hou, J. et al. Electrical Behavior and Electron Transfer Modulation of Nickel-Copper Nanoalloys Confined in Nickel-Copper Nitrides Nanowires Array Encapsulated in Nitrogen-Doped Carbon Framework as Robust Bifunctional Electrocatalyst for Overall Water Splitting. *Adv. Funct. Mater.* **28**, 1803278 (2018).
- (3) Yoon, T. & Kim, K. S. One-Step Synthesis of CoS-Doped β -Co(OH)₂@Amorphous MoS_{2+x} Hybrid Catalyst Grown on Nickel Foam for High-Performance Electrochemical Overall Water Splitting. *Adv. Funct. Mater.* **26**, 7386-7393 (2016).
- (4) Hou, J., Wu, Y., Cao, S., Sun, Y. & Sun, L. Active Sites Intercalated Ultrathin Carbon Sheath on Nanowire Arrays as Integrated Core-Shell Architecture: Highly Efficient and Durable Electrocatalysts for Overall Water Splitting. *Small* **13**, 1702018 (2017).
- (5) R, Smith. et al. Photochemical Route for Accessing Amorphous Metal Oxide Materials for Water Oxidation Catalysis. *Science* **340**, 60-63 (2013).

[4] Those measured current densities were all calculated with respect to the geometric areas of the assembled electrodes, not directly related to the intrinsic activities of these catalyst samples. In order to show whether the intrinsic activity is improved or not, turnover frequency (TOF) value of both HER and OER should be calculated and compared with those of other catalysts.

Response : Thanks for the comments and suggestions from the reviewer. For rough estimation per-site turnover frequency (TOF), we previously carried out according to the previously approach adopted by Jaramillo et al.¹ In this way, the geometric areas of the assembled electrodes were hypothesized. Then TOF values were calculated. In order to ascertain the reasonable TOF values, cycle voltammetry (CV) method can be applied in this part. The TOF values (s⁻¹) were calculated with the following formula:

$$\text{TOF} = \frac{I}{2nF}$$

I: current density extracted from the LSV curves;

F: Faraday constant;

n: the number of active sites.

Cycle voltammetry measurements were conducted between -0.2 V and 0.6 V vs. RHE in 1M PBS (pH=7) at a scan rate of 50 mV s⁻¹. The absolute components of the voltammetric charges tested during one CV cycle were calculated. Assuming a one-electron process for both reduction and oxidation, the absolute charges was divided by two and the Faraday constant to get the number of active sites of the catalysts. The upper limit of active sites for NiMoO_x/NiMoS could be calculated according to the equation: $n=Q/2F$.

Table R2. The turnover frequency (TOF) values for various HER electrocatalysts at different overpotentials in alkaline condition.

Catalysts	TOF values in HER		
	$\eta=50$ mV	$\eta=100$ mV	$\eta=200$ mV
NiMoO _x /NiMoS	0.28 s ⁻¹	1.97 s ⁻¹	10.08 s ⁻¹
NiO _x /Ni ₃ S ₂	0.16 s ⁻¹	0.59 s ⁻¹	3.80 s ⁻¹
MoO _x /MoS ₂	0.007 s ⁻¹	0.027 s ⁻¹	0.35 s ⁻¹
NiMoS	0.004 s ⁻¹	0.0435 s ⁻¹	0.13 s ⁻¹

Table R3. The turnover frequency (TOF) values for various OER electrocatalysts at different overpotentials in alkaline condition.

Catalysts	TOF values in OER	
	$\eta=250$ mV	$\eta=300$ mV
NiMoO _x /NiMoS	3.82 s ⁻¹	9.93 s ⁻¹
NiO _x /Ni ₃ S ₂	1.55 s ⁻¹	4.78 s ⁻¹
MoO _x /MoS ₂	0.078 s ⁻¹	0.56 s ⁻¹

To explore the intrinsic electrocatalytic performance of each active sites, the turnover frequency (TOF) is calculated. Based on above-mentioned analysis, cycle voltammetry approach is regarded to the promising way to determine the reasonable results. The TOF value of NiMoO_x/NiMoS array (1.97 s⁻¹) at the overpotential of 100 mV is ~45 times higher than that of NiMoS array (0.0435 s⁻¹) for HER, as shown in Table R2. In addition, the TOF value of NiMoO_x/NiMoS array at the overpotential of 100 mV is higher than that of NiO_x/Ni₃S₂ and MoO_x/MoS₂ arrays for OER, as shown in Table R3. Meanwhile, the comparison of TOF values of HER and OER catalysts in alkaline condition has been presented in Table R4 and Table R5.

Table R4. Comparison of TOF values of HER catalysts in alkaline condition.

Catalysts	TOF (H ₂ s ⁻¹ @mV)	References
NiCo ₂ P _x	0.056 s ⁻¹ at 100 mV	[2]
MoNi ₄ /MoO _{3-x}	1.13 s ⁻¹ at 100 mV	[3]
N-NiCo ₂ S ₄	1.0 s ⁻¹ at 125 mV	[4]
NiO@1T MoS ₂	0.7 s ⁻¹ at 130 mV	[5]
Mo ₁ N ₁ C ₂	1.46 s ⁻¹ at 150 mV	[6]
Co-NiS ₂	4.1 s ⁻¹ at 200 mV	[7]
MoO ₃ @MoS ₂	1.93 s ⁻¹ at 250 mV	[8]
NiMoO _x /NiMoS	0.28 s ⁻¹ at 50 mV	this work
NiMoO _x /NiMoS	1.97 s ⁻¹ at 100 mV	this work

Table R5. Comparison of TOF values of OER catalysts in alkaline condition.

Catalysts	TOF (H ₂ S ⁻¹ @mV)	References
Ir/Ni(OH) ₂	3.8 s ⁻¹ at 300 mV	[9]
e-ICLDH@GDY/NF	2.34 s ⁻¹ at 250 mV	[10]
FeCo-MOF-EH	0.062 s ⁻¹ at 300 mV	[11]
Ni-NHGF	0.72 s ⁻¹ at 300 mV	[12]
Ir-NiO	1.37 s ⁻¹ at 300 mV	[13]
CoSe ₂ -D _{Fe} -V _{Co}	0.045 s ⁻¹ at 280 mV	[14]
NiMoO _x /NiMoS	3.82 s ⁻¹ at 250 mV	this work

References:

(1) Benck, J. D. *et al.* Amorphous Molybdenum Sulfide Catalysts for Electrochemical Hydrogen Production: Insights into the Origin of Their Catalytic Activity. *ACS Catal.* **2**,

1916-1923 (2012).

- (2) Zhang, R. et al. Ternary NiCo₂P_x Nanowires as pH-Universal Electrocatalysts for Highly Efficient Hydrogen Evolution Reaction. *Adv. Mater.* **29**, 1605502 (2017).
- (3) Chen, Y. Y. et al. Self-Templated Fabrication of MoNi₄/MoO_{3-x} Nanorod Arrays with Dual Active Components for Highly Efficient Hydrogen Evolution. *Adv. Mater.* **29**, 1703311 (2017).
- (4) Wu, Y. et al. Electron Density Modulation of NiCo₂S₄ Nanowires by Nitrogen Incorporation for Highly Efficient Hydrogen Evolution Catalysis. *Nat. Commun.* **9**, 1425 (2018).
- (5) Huang, Y. et al. Atomically Engineering Activation Sites onto Metallic 1T-MoS₂ Catalysts for Enhanced Electrochemical Hydrogen Evolution. *Nat. Commun.* **10**, 982 (2019).
- (6) Chen, W. et al. Rational Design of Single Molybdenum Atoms Anchored on N-Doped Carbon for Effective Hydrogen Evolution Reaction. *Angew. Chem. Int. Ed.* **56**, 16086-16090 (2017).
- (7) Yin, J. et al. Atomic Arrangement in Metal-Doped NiS₂ Boosts the Hydrogen Evolution Reaction in Alkaline Media. *Angew. Chem. Int. Ed.* **58**, 18676-18682 (2019).
- (8) Huang, L. B. et al. Self-Limited on-Site Conversion of MoO₃ Nanodots into Vertically Aligned Ultrasmall Monolayer MoS₂ for Efficient Hydrogen Evolution. *Adv. Energy Mater.* **8**, 1800734 (2018).
- (9) Hui, L. et al. Overall Water Splitting by Graphdiyne-Exfoliated and -Sandwiched Layered Double-Hydroxide Nanosheet Arrays. *Nat. Commun.* **9**, 5309 (2018).
- (10) Dou, Y. et al. Approaching the Activity Limit of CoSe₂ for Oxygen Evolution via Fe doping and Co Vacancy. *Nat. Commun.* **11**, 1664 (2020).
- (11) Zhao, G., Li, P., Cheng, N., Dou, S. X. & Sun, W. An Ir/Ni(OH)₂ Heterostructured Electrocatalyst for the Oxygen Evolution Reaction: Breaking the Scaling Relation, Stabilizing Iridium(V), and Beyond. *Adv. Mater.* **32**, 2000872 (2020).
- (12) Tian, J., Jiang, F., Yuan, D., Zhang, L., Chen, Q. & Hong, M. Electric-Field Assisted In Situ Hydrolysis of Bulk Metal-Organic Frameworks (MOFs) into Ultrathin Metal Oxyhydroxide Nanosheets for Efficient Oxygen Evolution. *Angew. Chem. Int. Ed.* **59**, 13101-13108 (2020).
- (13) Fei, H. et al. General Synthesis and Definitive Structural Identification of MN₄C₄ Single-Atom Catalysts with Tunable Electrocatalytic Activities. *Nat. Catal.* **1**, 63 (2018).
- (14) Wang, Q. et al. Ultrahigh-Loading of Ir Single Atoms on NiO Matrix to Dramatically Enhance Oxygen Evolution Reaction. *J. Am. Chem. Soc.* **142**, 7425-7433 (2020).

[5] In the manuscript, it is very impressive that the very large catalytic current densities for both HER and OER are driven by the low potentials. In this case, the Faraday efficiencies of OER, HER and overall water splitting should be given, confirming the real electrocatalytic activities of NiMoO_x/NiMoS array.

Response : Thanks for the comments and suggestions from the reviewer. It is very impressive that the very large catalytic current densities for both HER and OER are driven by the low potentials. In this case, the Faraday efficiencies of OER, HER and overall water splitting have been presented, confirming the real electrocatalytic activities of NiMoO_x/NiMoS array. To determination of Faradaic efficiency, the Faradic efficiency of HER or OER catalysts is defined as the ratio of the amount of experimentally determined hydrogen or oxygen to that of the theoretically expected hydrogen or oxygen from the HER or OER reaction in 1 M KOH aqueous solution by use of an online gas chromatography system (GC, Techcomp GC 7890T, Ar carrier gas, Thermo Conductivity Detector). As for the theoretical value, we assumed that 100% current efficiency during the reaction, which means only the HER or OER process was occurring at the working electrode. The theoretically expected amount of hydrogen or oxygen was then calculated by applying the Faraday law, which states that the passage of 96485.4 C causes 1 equivalent of reaction. Afterwards, the amount of hydrogen evolution of NiMoO_x/NiMoS array was measured in comparison of theoretical quantity in Figure R4, presenting a promising Faraday efficiency of 99.6±0.3% towards real water splitting into hydrogen. Typically, the amount of oxygen evolution of NiMoO_x/NiMoS array was measured in comparison of theoretical quantity in Figure R5, presenting OER Faraday efficiency of 97.5±0.4% owe to the synergistic effect of the morphology, defect and heterostructure engineering.

Figure R4. The yields of hydrogen theoretically calculated from HER current and tested from an online gas chromatography system by use of as-prepared NiMoO_x/NiMoS heterostructure array at a current density of 10 mA cm⁻² in 1 M KOH solution.

Figure R5. The yields of hydrogen theoretically calculated from OER current and tested from an online gas chromatography system by use of as-prepared NiMoO_x/NiMoS heterostructure array at a current density of 10 mA cm⁻² in 1 M KOH solution.

The Faraday efficiencies of NiMoO_x/NiMoS array for the HER and OER were measured quantitatively from the total amount of charge passed through the cell during electrolysis and the total amounts of evolved gas were recorded by an online gas chromatography system. The amount of experimentally generated H₂ and O₂ matches well with the theoretically calculated amount under the total charge during the electrolysis process in Figure R6, suggested that the FEs are close to 99.6±0.3% and 97.3±0.5% for the HER and OER, with the ratio of H₂ and O₂ being close to 2 : 1.

Figure R6. The yields of hydrogen and oxygen theoretically calculated from HER and OER currents towards overall water splitting and tested from an online gas chromatography system by use of as-prepared NiMoO_x/NiMoS heterostructure array at a current density of 10 mA cm⁻² in 1 M KOH solution.

[6] The intrinsic activity should play an important role in electrocatalysis, so it is more valuable to take this into account, in-depth understanding the real electrocatalytic process in this work.

Response : Thanks for the comments and suggestions from the reviewer. To explore the intrinsic electrocatalytic HER performance of each active sites, the turnover frequency (TOF) is calculated in Table R2. The TOF value of NiMoO_x/NiMoS array (1.97 s⁻¹) at the overpotential of 100 mV is ~45 times higher than that of NiMoS array (0.0435 s⁻¹). Afterwards, the amount of hydrogen evolution of NiMoO_x/NiMoS array was measured in comparison of theoretical quantity in Figure R4, presenting a promising Faraday efficiency of 99.6±0.3% towards real water splitting into hydrogen. Typically, the amount of oxygen evolution of NiMoO_x/NiMoS array was measured in comparison of theoretical quantity in Figure R5, presenting OER Faraday efficiency of 97.5±0.4% owe to the synergistic effect of the morphology, defect and heterostructure engineering. Especially, the Faraday efficiencies of NiMoO_x/NiMoS array for the HER and OER were measured quantitatively from the total amount of charge passed through the cell during electrolysis and the total amounts of evolved gas were recorded by an online gas chromatography system. The amount of experimentally generated H₂ and O₂ matches well with the theoretically calculated amount under the total charge during the electrolysis process in Figure R6, suggested that the FEs are close to 99.6±0.3% and 97.3±0.5% for the HER and OER, with the ratio of H₂ and O₂ being close to 2 : 1.

[7] There are some reports about the electrocatalysts, delivering large catalytic current densities (e.g., 500 and 1000 mA cm⁻²) for practical application. Can the authors make a comparison among the catalytic activities of this catalyst in the manuscript and these reported catalysts?

Response : Thanks for the comments and suggestions from the reviewer. In comparison of different electrocatalysts, such as CoFeZr oxides,¹ MoS₂/Ni₃S₂,² MoS₂/Co₉S₈/Ni₃S₂,³ O-CoMoS,⁴ NC-NiCu-NiCuN,⁵ and commercial Pt/C||IrO₂ electrodes (Figure R1), the lower voltage at 10 mA cm⁻² was obtained by NiMoO_x/NiMoS array. Typically, the record low voltages of 1.60 and 1.66 V of two-electrode system in 6 M KOH solution at 60 °C were achieved for the industrial current densities of 500 and 1000 mA cm⁻², respectively, and it is still better than that of Pt/C||IrO₂ couple. In comparison of overall water splitting activities of bifunctional electrocatalysts (Table R6), NiMoO_x/NiMoS array presented the best electrocatalytic performance.

Table R6. Comparison of overall water splitting activities of bifunctional electrocatalysts.

Catalysts	Potential at 500 mA cm ⁻²	References
FeP/Ni ₂ P	1.72 V	[6]
RFNOH-10 NiFe-LDH/NF	1.69 V	[7]
Cu@NiFe LDH Ni _{2(1-x)} Mo _{2x} P	1.82 V	[8]
Ni NWs Ni _{0.8} Fe _{0.2} -AHNA	1.702 V	[9]
MoNi ₄ /SSWISSW Rs-12h	1.978 V	[10]
NFN-MOF/NF	1.9 V	[11]
NiMoO _x /NiMoS NiMoO _x /NiMoS	1.6 V	this work

References:

- (1) Huang, L. et al. Zirconium-Regulation-Induced Bifunctionality in 3D Cobalt-Iron Oxide Nanosheets for Overall Water Splitting. *Adv. Mater.* **31**, 1901439 (2019).
- (2) Zhang, J. et al. Interface Engineering of MoS₂/Ni₃S₂ Heterostructures for Highly Enhanced Electrochemical Overall-Water-Splitting Activity. *Angew. Chem. Int. Ed.* **55**, 6702-6707 (2016).
- (3) Yang, Y. et al. Hierarchical Nanoassembly of MoS₂/Co₉S₈/Ni₃S₂/Ni as a Highly Efficient Electrocatalyst for Overall Water Splitting in a Wide pH Range. *J. Am. Chem. Soc.* **141**, 10417-10430 (2019).
- (4) Feng, J. et al. Efficient Hydrogen Evolution on Cu Nanodots-Decorated Ni₃S₂ Nanotubes by Optimizing Atomic Hydrogen Adsorption and Desorption. *J. Am. Chem. Soc.* **140**, 610 (2018).
- (5) Hou, J. et al. Electrical Behavior and Electron Transfer Modulation of Nickel-Copper Nanoalloys Confined in Nickel-Copper Nitrides Nanowires Array Encapsulated in Nitrogen-Doped Carbon Framework As Robust Bifunctional Electrocatalyst for Overall Water Splitting. *Adv. Funct. Mater.* **28**, 1803278 (2018).
- (6) Yu, F. et al. High-Performance Bifunctional Porous Non-Noble Metal Phosphide Catalyst for Overall Water Splitting. *Nat. Commun.* **9**, 2551 (2018).
- (7) Xiao, X. et al. In Situ Growth of Ru Nanoparticles on (Fe,Ni)(OH)₂ to Boost Hydrogen Evolution Activity at High Current Density in Alkaline Media. *Small Methods* **4**, 1900796 (2020).
- (8) Yu, L. et al. Ternary Ni_{2(1-x)}Mo_{2x}P Nanowire Arrays toward Efficient And Stable Hydrogen Evolution Electrocatalysis under Large-Current-Density. *Nano Energy* **53**, 492-500 (2018).
- (9) Liang, C. et al. Exceptional Performance of Hierarchical Ni-Fe Oxyhydroxide@NiFe Alloy Nanowire Array Electrocatalysts for Large Current Density Water Splitting. *Energy*

Environ. Sci. **13**, 86-95 (2020).

(10) Jothi, V. R., Karuppasamy, K., Maiyalagan, T., Rajan, H., Jung, C. Y. & Yi, S. C. Corrosion and Alloy Engineering in Rational Design of High Current Density Electrodes for Efficient Water Splitting. *Adv. Energy Mater.* **10**, 1904020 (2020).

(11) Senthil, Raja. D., Chuah, X. F. & Lu, S. Y. In Situ Grown Bimetallic MOF-Based Composite as Highly Efficient Bifunctional Electrocatalyst for Overall Water Splitting with Ultrastability at High Current Densities. *Adv. Energy Mater.* **8**, 1801065 (2018).

[8] The 'Discussion' section looks like 'Conclusions' in this work. The authors need to comprehensively discuss on their study and new findings if the results and discussion can be split in this manuscript.

Response : Thanks for the comments and suggestions from the reviewer. In conclusion, hierarchical defective transition bimetal oxides/sulfides array supported on nickel foam was fabricated by novel oxidation/hydrogenation-induced surface reconfiguration strategies by use of NiMoS architectures as the precursor, interacting two-dimensional MoO_x/MoS₂ nanosheets attached on one-dimensional NiO_x/Ni₃S₂ nanorods array. To optimize the electrocatalytic performance, the influence of oxygen plasma power and hydrogenation temperature upon HER and OER of NiMoO_x/NiMoS array was conducted, corresponding the best plasma power of 100 W and thermal treatment temperature at 400 °C of NiMoO_x/NiMoS array. Benefiting from defect and heterostructure engineering, as-synthesized NiMoO_x/NiMoS heterostructure array presented the remarkable electrocatalytic performance in alkaline condition, achieving low overpotentials of 38, 89, 174 and 236 mV for HER and 186, 225, 278 and 334 mV for OER at 10, 100, 500 and 1000 mA cm⁻², even surviving at large current density of 100 and 500 mA cm⁻² with long-term stability in 1 M KOH solution at 25 °C. For the first time, the defective transition bimetal oxides/sulfides heterostructure array as the industrially promising electrocatalyst is reported as the typical model. The remarkable electrocatalytic performance is ascribed to not only the simultaneous modulation of component and geometric structure, but also the systematic optimization of charge transfer, abundant electrocatalytic active sites and exceptionally synergistic effect of heterostructure interfaces. Density functional theory calculations reveal that the coupling interface between NiMoO_x and NiMoS facilitates adsorption energies and accelerates water splitting kinetics, thus promoting the catalytic performance. Especially, the assembled two-electrode cell by use of NiMoO_x/NiMoS array delivered the industrially required current densities of 500 and 1000 mA cm⁻² at record low cell voltages of 1.60 and 1.66 V in 6 M KOH solution at 60 °C, along with excellent durability, outperforming most of transition metal-based bifunctional electrocatalysts reported to date. Given hierarchical transition heterostructures array as

typical model, this work paves avenues to the development of excellent electrocatalysts by interface atomic activation sites engineering for large-scale energy conversion applications.

Reviewer #2 (Remarks to the Author):

In this manuscript, hierarchical transition bimetal oxides/sulfides arrays (NiMoO_x/NiMoS) were fabricated by an oxidation/hydrogenation-induced surface reconfiguration strategy and investigated for water splitting. A suite of electrochemical analysis and characterizations clearly indicate the formation of 3D, defect-rich heterostructure. Impressive electrocatalytic performance for HER, OER and overall water splitting were demonstrated by thorough electrochemical tests. Active sites and reaction mechanism were further revealed via DFT studies. This work provides an attractive strategy of morphology, defect and heterostructure engineering upon hierarchical oxide/sulfides for water splitting, and is thus publishable. However, there are also several questions to be addressed.

We are grateful to the reviewer's comments. We have made careful revisions according to each comment, as summarized below.

[1] In introduction section, the shortcomings and remaining problems of reported catalysts are suggested to highlight. In addition to enumerate relevant materials, the basis and reason for the design and selection of the catalyst and the structure should be explained more clearly.

Response : Thanks for the comments and suggestions from the reviewer. With regard to transition metal dichalcogenides, MoS₂ and Ni₃S₂ materials have been extensively explored as HER electrocatalysts.¹⁻³ However, the HER performance of transition metal sulfides is limited by poor charge transport, low active site reactivity and inefficient electrical contact with the supported catalysts.⁴ Especially, the generation of S-H_{ads} bonds (H atoms adsorption, H_{ads}) on the surface of metal sulfides is beneficial for H adsorption, while it is difficult to conduct the conversion of the H_{ads} to H₂.^{5,6} Moreover, the OER performance of metal sulfides remains far from satisfactory.⁷⁻¹¹ To further enhance the performance of the electrocatalysts, defect engineering is an effective way to regulate reaction kinetics. For example, vacancies confined in MoSe₂ nanosheets,¹² Fe vacancies in δ-FeOOH nanosheets,¹³ sulfur vacancies in MoS₂,¹⁴ defective δ-MnO₂ nanosheets,¹⁵ perovskite hydroxide with vacancies, Co₃O₄ with oxygen vacancies and defect-rich carbon,¹⁶⁻²² have been synthesized for steering OER and HER performance. However, it is elusive to engineer the atomic activation sites of the catalysts at the interfaces, limiting the activities for electrocatalytic water splitting. Owe to long-time durability as major

obstacles, there is less report about the electrocatalysts, delivering large catalytic current densities (e.g., 500 and 1000 mA cm⁻²) for practical application.²⁰⁻²² Based above-mentioned analysis, it is essential to design the rational heterostructures through the combined regulation of morphology, defect and heterostructure, engineering interface atomic activation sites, optimizing energy adsorption and accelerating water splitting kinetics towards large-scale electrolysis. In introduction section, the shortcomings and remaining problems of reported catalysts have been added and highlighted. In addition to enumerate relevant materials, the basis and reason for the design and selection of the catalyst and the structure have been explained in introduction section.

References:

- (1) Wang, X. et al. Single-Atom Vacancy Defect to Trigger High-Efficiency Hydrogen Evolution of MoS₂. *J. Am. Chem. Soc.* 142, 4298 (2020).
- (2) Luo, Z. et al. Reactant Friendly Hydrogen Evolution Interface Based on Di-anionic MoS₂ Surface. *Nat. Commun.* 11, 1116 (2020).
- (3) He, W. et al. Fluorine-Anion-Modulated Electron Structure of Nickel Sulfide Nanosheet Arrays for Alkaline Hydrogen Evolution. *ACS Energy Lett.* 4, 2905 (2019).
- (4) Feng, L. et al. High-Index Faceted Ni₃S₂ Nanosheet Arrays as Highly Active and Ultrastable Electrocatalysts for Water Splitting. *J. Am. Chem. Soc.* 137, 14023 (2015).
- (5) Zhu, H. et al. When Cubic Cobalt Sulfide Meets Layered Molybdenum Disulfide: A Core-Shell System toward Synergetic Electrocatalytic Water Splitting. *Adv. Mater.* 27, 4752 (2015).
- (6) Feng, J. et al. Efficient Hydrogen Evolution on Cu Nanodots-Decorated Ni₃S₂ Nanotubes by Optimizing Atomic Hydrogen Adsorption And Desorption. *J. Am. Chem. Soc.* 140, 610 (2018).
- (7) Wu, Y. et al. Coupling Interface Constructions of MoS₂/Fe₅Ni₄S₈ Heterostructures for Efficient Electrochemical Water Splitting. *Adv. Mater.* 30, 1803151 (2018).
- (8) Zhang, J. et al. Interface Engineering of MoS₂/Ni₃S₂ Heterostructures for Highly Enhanced Electrochemical Overall-Water-Splitting Activity. *Angew. Chem. Int. Ed.* 55, 6702-6707 (2016).
- (9) An, T. et al. Interlaced NiS₂-MoS₂ Nanoflake-Nanowires as Efficient Hydrogen Evolution Electrocatalysts in Basic Solutions. *J. Mater. Chem. A* 4, 13439-13443 (2016).
- (10) Yang, Y. et al. Hierarchical Nanoassembly of MoS₂/Co₉S₈/Ni₃S₂/Ni as a Highly Efficient Electrocatalyst for Overall Water Splitting in a Wide pH Range. *J. Am. Chem. Soc.* 141, 10417-10430 (2019).
- (11) Li, H. et al. Systematic Design of Superaerophobic Nanotube-Array Electrode

Comprised of Transition-Metal Sulfides for Overall Water Splitting. *Nat. Commun.* 9, 2452 (2018).

(12) Gao, D. et al. Dual-Native Vacancy Activated Basal Plane and Conductivity of MoSe_2 with High-Efficiency Hydrogen Evolution Reaction. *Small* 14, 1704150 (2018).

(13) Liu, B. et al. Iron Vacancies Induced Bifunctionality in Ultrathin Ferrous Hydroxide Nanosheets for Overall Water Splitting. *Adv. Mater.* 30, 1803144 (2018).

(14) Yin, Y. et al. Contributions of Phase, Sulfur Vacancies, and Edges to the Hydrogen Evolution Reaction Catalytic Activity of Porous Molybdenum Disulfide Nanosheets. *J. Am. Chem. Soc.* 138, 7965-7972 (2016).

(15) Zhao, Y. et al. Defect-Engineered Ultrathin $\delta\text{-MnO}_2$ Nanosheet Arrays as Bifunctional Electrodes for Efficient Overall Water Splitting. *Adv. Energy Mater.* 7, 1700005 (2017).

(16) Chen, D. et al. Preferential Cation Vacancies in Perovskite Hydroxide for The Oxygen Evolution Reaction. *Angew. Chem. Int. Ed.* 57, 8691-8696 (2018).

(17) Xiao, Z. et al. Filling the Oxygen Vacancies in Co_3O_4 with Phosphorus: An Ultra-Efficient Electrocatalyst for Overall Water Splitting. *Energy Environ. Sci.* 10, 2563-2569 (2017).

(18) Wang, Y. et al. 3D Carbon Electrocatalysts in Situ Constructed by Defect-Rich Nanosheets and Polyhedrons from NaCl-Sealed Zeolitic Imidazolate Frameworks. *Adv. Funct. Mater.* 28, 1705356 (2018).

(19) R, Smith. et al. Photochemical Route for Accessing Amorphous Metal Oxide Materials for Water Oxidation Catalysis. *Science* 340, 60-63 (2013).

(20) Liu, Y. et al. Corrosion Engineering towards Efficient Oxygen Evolution Electrodes with Stable Catalytic Activity for Over 6000 Hours. *Nat. Commun.* 9, 2609 (2018).

(21) Zhang, J. et al. Modulation of Inverse Spinel Fe_3O_4 by Phosphorus Doping as an Industrially Promising Electrocatalyst for Hydrogen Evolution. *Adv. Mater.* 31, 1905107 (2019).

(22) Yu, L. et al. Non-Noble Metal-Nitride Based Electrocatalysts for High-Performance Alkaline Seawater Electrolysis. *Nat. Commun.* 10, 5106 (2019).

[2] Some results of the experimental characterization need to be checked. For example:

(1) For the results of TEM analysis on defect structures marked in Figure 3e, the size of each white dot looks much bigger than the size of an atom. Are these dots (supposed to be defects) attributed to electron beam irradiation damage or derived from the material/sample itself?

Response : Thanks for the comments and suggestions from the reviewer. With regard to the analysis of TEM image in Figure 3e, these dots are attributed to electron beam irradiation damage. The defects with yellow circles have been deleted, avoiding the misunderstanding between electron beam irradiation damage and the defects.

(2) Please check the lattice fringes for Ni_3S_2 marked in Figure 3e. The fringes are horizontal to the edge but interlayer distance is marked vertical.

Response : Thanks for the comments and suggestions from the reviewer. The lattice fringes for Ni_3S_2 has been marked in Figure 3e as below. With regard to $\text{NiO}_x/\text{Ni}_3\text{S}_2$ species from hierarchical defective $\text{NiMoO}_x/\text{NiMoS}$ heterostructure array, high-angle annular dark-field scanning transmission electron microscope (HAADF-STEM) and the corresponding EDX elemental mapping have been conducted (Figure R7 and Figure R8), indicating the formation of ultrathin NiO_x layer on the Ni_3S_2 . Combined with the analysis of TEM, HAADF-STEM and the elemental mapping, $\text{NiO}_x/\text{Ni}_3\text{S}_2$ species have been produced in hierarchical defective $\text{NiMoO}_x/\text{NiMoS}$ heterostructure array.

Figure R7. TEM image of $\text{NiO}_x/\text{Ni}_3\text{S}_2$ species from hierarchical defective $\text{NiMoO}_x/\text{NiMoS}$ heterostructure array. Scale bar, 5 nm.

Figure R8. HAADF-STEM image and the corresponding EDX elemental mapping of $\text{NiO}_x/\text{Ni}_3\text{S}_2$ species from hierarchical defective $\text{NiMoO}_x/\text{NiMoS}$ heterostructure array. Scale bar, 10 nm.

(3) The images in Figure S5 are identical to that in Figure 3f-i. It's not necessary to repeat this information.

Response : Thanks for the comments and suggestions from the reviewer. Figure S5 has been deleted in supporting information.

[3] In the part of theoretical calculation, the additional explanations for the established catalyst models and the exploration of atomic activation sites at the interface need to be supplemented.

(1) The modeling of metal oxides is described as "used O atom to replace the S atom on the edge of MoS_2 and the surface of Ni_3S_2 and the interface of MoS_2 and Ni_3S_2 , respectively. To simulate the edge, surface and interface incorporate with oxides how to effects on the catalyst overall water splitting". The crystal facets of MoO_2 and MoO_3 can be observed from TEM, and the diffraction peaks of MoO_2 , MoO_3 and NiO can be obtained from XRD results. The crystal structure information of metal oxides can be

confirmed by the above results, but the theoretical models of NiO_x , MoO_x and NiMoO_x are represented by an O atom instead of S atom in Ni_3S_2 , MoS_2 and NiMoS , respectively, which might not embody "multi-interfaces of dual-metal based oxide-sulfide heterostructures" claimed in the article. Please rethink the deviations between the theoretical models and structures characterized in experiments.

Response : Thanks for the comments and suggestions from the reviewer. Our slab models at begin is built a $\text{MoO}_2/\text{MoS}_2/\text{Ni}_3\text{S}_2/\text{NiO}$ nanoribbon to simulate multi-interfaces. Unfortunately, due to the large than 10% lattice mismatch between MoO_2 and MoS_2 which induced huge lattice stress and monolayer Ni_3S_2 cannot keep the bulk phase structures result in seriously structure distortion. Meanwhile, considering the limitations of computing resources, so we adopt the previous literature proposed methods.^{1,2} For $\text{NiO}_x/\text{Ni}_3\text{S}_2$ and $\text{MoO}_x/\text{MoS}_2$ was belong to monometallic heterostructure, so we use the O substitute doped the MoS_2 (002) ribbon and Ni_3S_2 (10-1) surface to represent the $\text{NiO}_x/\text{Ni}_3\text{S}_2$ and $\text{MoO}_x/\text{MoS}_2$ interface models. For $\text{Ni}_3\text{S}_2/\text{MoS}_2$ interface was built by MoS_2 (002) ribbon normal to the S-terminated Ni_3S_2 (-101) slab to simulate the interface between MoS_2 (002) and Ni_3S_2 (101) surface. The lattice mismatch of Ni_3S_2 and MoS_2 is about 3% by compressing MoS_2 monolayer. Equally, we use O atom substitute doped $\text{Ni}_3\text{S}_2/\text{MoS}_2$ interface to represent the $\text{NiMoO}_x/\text{NiMoS}$ interface. Finally, our slab model including three interfaces, that is, $\text{NiO}_x/\text{Ni}_3\text{S}_2$, $\text{MoO}_x/\text{MoS}_2$ and $\text{NiMoO}_x/\text{NiMoS}$ interfaces. Therefore, our slab model reflects the feature of multi-interfaces of dual-metal based oxide-sulfide heterostructures.

References:

- (1) Peng, L. et al. Rationally design of monometallic $\text{NiO-Ni}_3\text{S}_2/\text{NF}$ heteronanosheets as bifunctional electrocatalysts for overall water splitting. *J. Catal.* **369**, 345-351 (2019).
- (2) Yang, Y. et al. Hierarchical Nanoassembly of $\text{MoS}_2/\text{Co}_9\text{S}_8/\text{Ni}_3\text{S}_2/\text{Ni}$ as a Highly Efficient Electrocatalyst for Overall Water Splitting in a Wide pH Range. *J. Am. Chem. Soc.* **141**, 10417-10430 (2019).

(2) The manuscript proposed that the constructed interfaces among various heterostructures facilitate the charge transfer and bring exceptionally synergistic effect by oxidation/hydrogenation-induced surface reconfiguration. However, the charge transfer of $\text{NiMoO}_x/\text{NiMoS}$ in the catalytic process has not been given from the theoretical calculation point of view, so it would be better to make a further explanation.

Response : Thanks for the comments and suggestions from the reviewer. In order to understand the charge transfer between the NiMoO_x/NiMoS interface, we calculated the electron charge difference by using the NiMoO_x/NiMoS heterostructure electron density subtracting the electron density of an individual MoS₂ and Ni₃S₂ structure (atoms are keep the same coordinates with heterostructures). The plane-averaged charge density difference along the Z direction was also calculated. It is clearly that a remarkable charge transfer and distribution occur in the interface region due to the formation of Mo-S bonds across this interface. The accumulation of electrons at interface would facilitate the fast electron transfer during the electrochemical catalysis process. The Bader charge analysis also show that during the HER and OER process the significant electron transfer between adsorption intermediates and NiMoO_x/NiMoS heterostructure. Therefore, the excellent HER and OER activities can be attributed to the electron transfer at the NiMoO_x/NiMoS interfaces.

Figure R9. (a) The charge density difference and (b) the plane-averaged charge density difference $\Delta\rho_z$ along the Z direction of NiMoO_x/NiMoS. The purple and orange regions represent the charge accumulation and depletion, respectively. The isosurface value was $0.005 \text{ e}/\text{\AA}^3$. The $\Delta\rho_z > 0$ represents the charge accumulation.

Figure R10. (a) and (b) are Bader charge analysis of H, OH, O and OOH intermediates adsorption on the interface of $\text{Ni}_3\text{S}_2/\text{MoS}_2$ and $\text{NiMoO}_x/\text{NiMoS}$, respectively. Here, the minus sign is the loss of electrons and plus sign means gaining electrons.

(3) The claim “the synergistic effect of the morphology, defect and heterostructure engineering in NiMoO_x/NiMoS array promote the generation of abundant active sites by engineering interface atomic activation sites, optimizing adsorption energies and accelerating water splitting kinetics” has been emphasized in the manuscript. However, the theoretical calculation does not reflect the characteristics of the defective structure, nor does it describe the influence of the defective structure on its catalytic performance.

Figure R11. Top view and side view of the adsorption configurations of H, OH, O and OOH intermediates on the (a, c) perfect monolayer MoS₂ and (b, d) monolayer MoS₂ with sulfur vacancy and corresponding free energy diagrams on the right panel. Dashed circles denote the sulfur vacancy site.

Response : Thanks for the comments and suggestions from the reviewer. In this work, the synergistic effect of the morphology, defect and heterostructure engineering in

NiMoO_x/NiMoS array promotes the generation of abundant active sites by engineering interface atomic activation sites, optimizing adsorption energies and accelerating water splitting kinetics. The influence of the morphology and the heterostructure upon the electrocatalytic activities has already been explored in this work. In order to investigate the influence of the defects, such as vacancies, upon the electrocatalytic performance, we calculate the Gibbs free energies of every step in HER and OER of perfect and with sulfur vacancy of monolayer MoS₂ (Figure R11). The calculated results show that the sulfur vacancies have ideal hydrogen absorption energy ($\Delta G_{H^+} = -0.07$ eV) compared to perfect monolayer MoS₂ ($\Delta G_{H^+} = 1.97$ eV), which is consistent with the previous reported literatures.^{1,2} However, for OER process, both perfect MoS₂ and MoS₂ with sulfur vacancy have poor catalysis performance.³ Since hydroperoxy intermediate is decomposed into O and OH intermediates adsorbed on the perfect MoS₂ monolayer (called *O*OH), and also decomposed at the sulfur vacancy site, O intermediate is adsorbed on the sulfur vacancy site, while OH intermediate is free above the surface (called *O(OH)).

References:

- (1) Li, H. et al. Activating and optimizing MoS₂ basal planes for hydrogen evolution through the formation of strained sulphur vacancies. *Nat. Mater.* **15**, 364 (2016).
- (2) Li, G. et al. All The Catalytic Active Sites of MoS₂ for Hydrogen Evolution. *J. Am. Chem. Soc.* **138**, 16632-16638 (2016).
- (3) German, E. & Gebauer, R. Why are MoS₂ monolayers not a good catalyst for the oxygen evolution reaction? *Appl. Surf. Sci.* **528**, 146591 (2020).
- (4) The O atoms in NiO_x, MoO_x, NiMoO_x are possible sites to involve in OER. If so, they may interact and is competitive with other active site, which would change the favored potential energy surface. Discussion along this line is suggested.

Response : Thanks for the comments and suggestions from the reviewer. For possible OER catalysis sites of NiO_x, MoO_x and NiMoO_x, we have performed computational screening (Figure R12~Figure R14). The calculation results show that the Ni atom in the NiMoO_x/NiMoS interface is the best OER catalytic active site. For NiO_x/Ni₃S₂ interface, the better OER catalytic active site is the S atom. For the pristine MoS₂, edge S atom is the mainly OER catalytic active site, whereas the MoO_x/MoS₂(S) interface are slightly increased the overpotential which is disadvantage for OER process. To sum up, the NiMoO_x/NiMoS interface is the mainly active sites for overall water splitting.

Figure R12. Possible OER sites for $\text{Ni}_3\text{S}_2/\text{MoS}_2$ (a,b) and corresponding free energy diagrams (c,d).

Figure R13. Possible OER sites for $\text{NiMoO}_x/\text{NiMoS}$ (a,b) and corresponding free energy diagrams (c,d).

Figure R14. Possible OER sites for (a) $\text{Ni}_3\text{S}_2(\text{S})$, (b, c) $\text{NiO}_x/\text{Ni}_3\text{S}_2(\text{S})$, (d) $\text{Ni}_3\text{S}_2(\text{Ni})$ and (e, f) $\text{NiO}_x/\text{Ni}_3\text{S}_2(\text{Ni})$ and corresponding free energy diagrams.

Figure R15. Possible OER site for (a, b) $\text{MoS}_2(\text{S})$, (c-f) $\text{MoO}_x/\text{MoS}_2(\text{S})$, (g) $\text{MoS}_2(\text{Mo})$ and (h) $\text{MoO}_x/\text{MoS}_2(\text{Mo})$ and corresponding free energy diagrams.

Reviewer #3 (Remarks to the Author):

Authors presented a non-PGM catalyst for water splitting. I think the activities reported herein are quite high, but I have some major concerns. I suggest major revision and authors should address the following issues before the articles publication.

We are grateful to the reviewer's comments. We have made careful revisions according to each comment, as summarized below.

Please see the comments below:

[1] I am not convinced about the measurements of surface area; I suggest that authors should elaborate on it. What were the values for measured electroactive surface area, what were the geometric areas? Do they correspond reasonably with each other, considering the porosity of nickel foams?

Response : Thanks for the comments and suggestions from the reviewer. The geometric area for each electrocatalysts is 1cm^2 . Considering the porosity of nickel foam, we tested the specific capacitance of NF. As shown in Figure R16, the value of NF is 0.965 mF cm^{-2} . So, the measured electroactive surface area is 23.1, 11.5, 9.9 and $7.2\text{ cm}^2_{\text{ECSA}}$ for $\text{NiMoO}_x/\text{NiMoS}$, $\text{NiO}_x/\text{Ni}_3\text{S}_2$, $\text{MoO}_x/\text{MoS}_2$ and NiMoS .

Figure R16. Electrochemical double-layer capacitance tests of nickel foam.

[2] Since the determination of ECSA in metal oxide catalysts is not precise, I suggest the authors report also the activities observed per mass, and compare the activities per mass with the literature mass activities.

Response : Thanks for the comments and suggestions from the reviewer. We have added the activities observed per mass, and compared the activities per mass with the literature mass activities.

Figure R17. Calculated mass activity of NiMoO_x/NiMoS for HER in 1 M KOH.

Table R7. Comparison of mass activity of HER electrocatalysts.

Catalysts	Mass activity (A g ⁻¹)	References
Ru@MWCNT	186 A g ⁻¹ at 20mV	[1]
Pt-CoS ₂ /CC	2.1 A g ⁻¹ at 100mV	[2]
R-MoS ₂ @NF	40 A g ⁻¹ at 100mV	[3]
Co-NiS ₂	100 A g ⁻¹ at 185mV	[4]
Pt@PCM	600 A g ⁻¹ at 185mV	[5]
NiCoN/C	204 A g ⁻¹ at 200mV	[6]
O _V -Co ₃ O ₄	2.12 A g ⁻¹ at 200mV	[7]
A-MoS ₂	40 A g ⁻¹ at 300mV	[3]
NiS ₂	150 A g ⁻¹ at 300mV	[4]
NiMoO _x /NiMoS	436 A g ⁻¹ at 200mV	this work

References:

(1) Kweon, D. H. et al. Ruthenium Anchored on Carbon Nanotube Electrocatalyst for Hydrogen Production with Enhanced Faradaic Efficiency. *Nat. Commun.* **11**, 1278 (2020).

- (2) Han, X. et al. Ultrafine Pt Nanoparticle-Decorated Pyrite-Type CoS₂ Nanosheet Arrays Coated on Carbon Cloth as a Bifunctional Electrode for Overall Water Splitting. *Adv. Energy Mater.* **8**, 1800935 (2018).
- (3) Anjum, M. A. R., Jeong, H. Y., Lee, M. H., Shin, H. S. & Lee, J. S. Efficient Hydrogen Evolution Reaction Catalysis in Alkaline Media by All-in-One MoS₂ with Multifunctional Active Sites. *Adv. Mater.* **30**, 1707105 (2018).
- (4) Yin, J. et al. Atomic Arrangement in Metal-Doped NiS₂ Boosts the Hydrogen Evolution Reaction in Alkaline Media. *Angew. Chem. Int. Ed.* **58**, 18676-18682 (2019).
- (5) Zhang, H. et al. Dynamic Traction of Lattice-Confined Platinum Atoms into Mesoporous Carbon Matrix for Hydrogen Evolution Reaction. *Sci. Adv.* **4**, eaao6657 (2018).
- (6) Lai, J., Huang, B., Chao, Y., Chen, X. & Guo, S. Strongly Coupled Nickel-Cobalt Nitrides/Carbon Hybrid Nanocages with Pt-Like Activity for Hydrogen Evolution Catalysis. *Adv Mater* **31**, 1805541 (2019).
- (7) Zhang, H., Zhang, J., Li, Y., Jiang, H., Jiang, H. & Li, C. Continuous oxygen vacancy engineering of the Co₃O₄ layer for an enhanced alkaline electrocatalytic hydrogen evolution reaction. *J. Mater. Chem. A* **7**, 13506-13510 (2019).

[3] Authors should report how many times the activities were reproduced and should include the errors bars to ensure the reproducibility of the results.

Response : Thanks for the comments and suggestions from the reviewer. We have conducted the activities for three times, corresponding the errors bars to ensure the reproducibility of the results. Afterwards, the amount of hydrogen evolution of NiMoO_x/NiMoS array was measured in comparison of theoretical quantity (Supplementary Fig. 12), presenting a promising Faraday efficiency of 99.6±0.3% towards real water splitting into hydrogen. Typically, the amount of oxygen evolution of NiMoO_x/NiMoS array was measured in comparison of theoretical quantity (Supplementary Fig. 15), presenting OER Faraday efficiency of 97.5±0.4% owe to the synergistic effect of the morphology, defect and heterostructure engineering.

[4] The Figure 7 shows the polarization curve for OER not the overall water splitting, please correct. Also please report where were the values for figure 7c were taken from, were they done in exact same conditions (electrolyte etc.).

Response : Thanks for the comments and suggestions from the reviewer. Figure 7 shows the polarization curve for overall water splitting. Especially, the electrocatalysis shown in Figure 7c was tested in 1 M KOH solution.

[5] I am wondering if they had any difficulties with bubble formation and blocking of the catalyst active site.

Response : Thanks for the comments and suggestions from the reviewer. To investigate the superaerophobic of as-prepared electrocatalysts, we performed contact angle measurement under aqueous solution. As shown in Fig. R18, the underwater bubble contact angle is 151.2° , 133.2° and 146.5° for $\text{NiMoO}_x/\text{NiMoS}$, $\text{NiO}_x/\text{Ni}_3\text{S}_2$ and $\text{MoO}_x/\text{MoS}_2$, respectively, indicating the hydrophobic properties. These properties demonstrated that the typical architecture could facilitate the release of the evolved gas bubbles and thus avoid the block of the catalyst active site.

Figure R18. Air-bubble contact angles under water for (a) $\text{NiMoO}_x/\text{NiMoS}$, (b) $\text{NiO}_x/\text{Ni}_3\text{S}_2$, (c) $\text{MoO}_x/\text{MoS}_2$. The bubble contact angles of $\text{NiMoO}_x/\text{NiMoS}$, $\text{NiO}_x/\text{Ni}_3\text{S}_2$ and $\text{MoO}_x/\text{MoS}_2$ were 151.2° , 133.2° and 146.5° , respectively.

References:

- (1) Xia, Z. & Guo, S. Strain Engineering of Metal-Based Nanomaterials for Energy Electrocatalysis. *Chem. Soc. Rev.* 48, 3265-3278 (2019).
- (2) Wei, C., Sun, S., Mandler, D., Wang, X., Qiao, S. Z. & Xu, Z. J. Approaches for Measuring The Surface Areas of Metal Oxide Electrocatalysts for Determining Their Intrinsic Electrocatalytic Activity. *Chem. Soc. Rev.* 48, 2518-2534 (2019).
- (3) Zou, X. & Zhang, Y. Noble Metal-Free Hydrogen Evolution Catalysts for Water Splitting. *Chem. Soc. Rev.* 44, 5148-5180 (2015).
- (4) Pi, Y. et al. Trimetallic Oxyhydroxide Coralloids for Efficient Oxygen Evolution Electrocatalysis. *Angew. Chem. Int. Ed.* 56, 4502-4506 (2017).
- (5) Huang, L. et al. Zirconium-Regulation-Induced Bifunctionality in 3D Cobalt-Iron Oxide Nanosheets for Overall Water Splitting. *Adv. Mater.* 31, 1901439 (2019).
- (6) Hao, S. et al. NiCoMo Hydroxide Nanosheet Arrays Synthesized via Chloride Corrosion for Overall Water Splitting. *ACS Energy Lett.* 4, 952-959 (2019).
- (7) Hou, J. et al. Active Sites Intercalated Ultrathin Carbon Sheath on Nanowire Arrays as

Integrated Core-Shell Architecture: Highly Efficient and Durable Electrocatalysts for Overall Water Splitting. *Small* 13, 1702018 (2017).

[6] On Figure 6d, what is meant by $100\text{mA}/\text{cm}^2$ at 225mV , does that potential correspond to the potential vs. RHE when $100\text{mA}/\text{cm}^2$ was observed? I have the same question for potential of $500\text{mA}/\text{cm}^2$ at 278mV .

Response : Thanks for the comments and suggestions from the reviewer. In comparison of NiMoS, $\text{MoO}_x/\text{MoS}_2$, $\text{NiO}_x/\text{Ni}_3\text{S}_2$, as-synthesized NiMoO_x/NiMoS array delivered the current densities of 10, 100, 500 and 1000 mA cm^{-2} at 0.038, 0.089, 0.174 and 0.236 V vs. RHE towards HER. In comparison of NiMoS, $\text{MoO}_x/\text{MoS}_2$, $\text{NiO}_x/\text{Ni}_3\text{S}_2$, as-synthesized NiMoO_x/NiMoS array delivered the current densities of 10, 100, 500 and 1000 mA cm^{-2} at 1.416, 1.455, 1.508 and 1.564 V vs. RHE towards OER, satisfying the requirements for commercial electrocatalytic application.

Figure R6. Chronoamperometric test of NiMoO_x/NiMoS array at current densities of 100 and 500 mA cm^{-2} towards HER and OER. Inset: polarization curves of NiMoO_x/NiMoS for the durability test after 2000 CV cycles.

[7] Figure 1 please label which colored ball corresponds to which element.

Response : Thanks for the comments and suggestions from the reviewer. We have labeled the colored ball, corresponding to the relative elements in Figure 1.

Figure R19. Schematic representation of synthesis and overall water splitting. (a) Synthesis illustration of hierarchical defective transition bimetal oxides/sulfides heterostructure array and (b) NiMoO_x/NiMoS array as two-electrode-cell towards large-scale electrolysis. Colored balls represent various elements (blue: Mo, pink: S, red: O, yellow: Ni).

[8] Figure 2 for the elemental mapping images, which image corresponds to which element please label.

Response : Thanks for the comments and suggestions from the reviewer. We added the element label in the mapping images (Figure R20).

Figure R20. Morphological and structural characterizations. SEM images of (a,d) MoS₂, (b,e) NiMoS, (c,f) NiMoO_x/NiMoS with the structure illustration (d); (g-j) elemental mapping images of NiMoO_x/NiMoS. Scale bar, (a-c) 5 μm; (d-f) 1 μm; (g-j) 10 μm.

[9] In overall the written English of the article needs major improvements. There are many minor mistakes such as on Page 15: NiMoO_x/NiMoS array was utilized as as cathode and anode in two-electrode...

Response : Thanks for the comments and suggestions from the reviewer. The whole manuscript has been checked, avoiding the mistakes in these statements.

REVIEWER COMMENTS

Reviewer #1 (Remarks to the Author):

The authors addressed my concerns. It is publishable now.

Reviewer #2 (Remarks to the Author):

From my side, I think the authors have sincerely and appropriately addressed all referees' comments and made satisfactory revision with an abundance of supplemented experimental and computational results and discussion. This quality of this updated version is largely improved to meet the standard of Nature Commun. I have no further concern on the manuscript.

Reviewer #3 (Remarks to the Author):

Unfortunately, after the revision the manuscript is not suitable for publication in Nature Commun., I recommend rejection.

I am not convinced about the accuracy of the electrochemical characterizations. There are many discrepancies, for example, authors claim they used 2x3cm (6cm²) Ni foam to prepare the catalyst but in the response letter they mention they have geometrical area of 1cm² catalyst, since the catalyst is merged into solution and both sides of Ni foam is immersed I would expect higher geometric area (if they used whole 6cm² sample and dipped it into electrolyte as they show in the SI, then geometric area will be 12cm²). The response to the comment about ECSA determination raised the following questions: what was the specific capacitance of Ni, NiMoOx/NiMoS and all the other catalysts used for finding the ECSA? Were they determined from the plots of current density vs the scan rate? What is the porosity of this Ni foam, for instance, the BET surface area? I suggest that the authors deposit their catalyst material on a flat surface rather than using a foam and then test the intrinsic activity, because the Ni foam will have much higher surface area than simple geometric area due to its porosity.

What was the mass loading of commercial Pt and IrO₂ catalysts onto Ni foam, how do they control this, how do they know the catalyst sticks to the Ni foam? Please measure the ECSA of Pt and IrO₂ catalysts from their respective CVs. Which company they were purchased from? Do they correspond well with their activities reported in the literature?

Moreover, authors didn't address my comments about the stability tests. For Figures 5d, 6d and 7d, they show the change in current density by holding at certain current density, this simply measures how stable is the potentiostat not the catalyst material. It is meaningful to show how the potential changes over time at constant current or how the current changes at constant potential over time. However, the plot of current density at constant current density over time is meaningless and doesn't provide any information about the catalyst material.

Moreover, authors claim atomic activation sites, defective structure of the catalyst but provide no experimental result to support this claim. Especially, after it was pointed out by the reviewer that defects on the TEM images are from beam and larger than single atomic defects. So I would suggest to rename, reconstruct the paper accordingly by avoiding terms such as atomic activation sites and defective structure if they don't provide any experimental evidence.

Editorial notes:

Reviewer comments & decisions:

REVIEWER COMMENTS

Reviewer #1 (Remarks to the Author):

The authors addressed my concerns. It is publishable now.

Reviewer #2 (Remarks to the Author):

From my side, I think the authors have sincerely and appropriately addressed all referees' comments and made satisfactory revision with an abundance of supplemented experimental and computational results and discussion. This quality of this updated version is largely improved to meet the standard of Nature Commun. I have no further concern on the manuscript.

Reviewer #3 (Remarks to the Author):

Unfortunately, after the revision the manuscript is not suitable for publication in Nature Commun., I recommend rejection.

I am not convinced about the accuracy of the electrochemical characterizations. There are many discrepancies, for example, authors claim they used 2x3cm (6cm²) Ni foam to prepare the catalyst but in the response letter they mention they have geometrical area of 1cm² catalyst, since the catalyst is merged into solution and both sides of Ni foam is immersed I would expect higher geometric area (if they used whole 6cm² sample and dipped it into electrolyte as they show in the SI, then geometric area will be 12cm²). The response to the comment about ECSA determination raised the following questions: what was the specific capacitance of Ni, NiMoO_x/NiMoS and all the other catalysts used for finding the ECSA? Were they determined from the plots of current density vs the scan rate? What is the porosity of this Ni foam, for instance, the BET surface area? I suggest that the authors deposit their catalyst material on a flat surface rather than using a foam and then test the intrinsic activity, because the Ni foam will have much higher surface area than simple geometric area due to its porosity.

What was the mass loading of commercial Pt and IrO₂ catalysts onto Ni foam, how do they control this, how do they know the catalyst sticks to the Ni foam? Please measure

the ECSA of Pt and IrO₂ catalysts from their respective CVs. Which company they were purchased from? Do they correspond well with their activities reported in the literature?

Moreover, authors didn't address my comments about the stability tests. For Figures 5d, 6d and 7d, they show the change in current density by holding at certain current density, this simply measures how stable is the potentiostat not the catalyst material. It is meaningful to show how the potential changes over time at constant current or how the current changes at constant potential over time. However, the plot of current density at constant current density over time is meaningless and doesn't provide any information about the catalyst material.

Moreover, authors claim atomic activation sites, defective structure of the catalyst but provide no experimental result to support this claim. Especially, after it was pointed out by the reviewer that defects on the TEM images are from beam and larger than single atomic defects. So I would suggest to rename, reconstruct the paper accordingly by avoiding terms such as atomic activation sites and defective structure if they don't provide any experimental evidence.

Responses to the reviewer's comments

Reviewer #1 (Remarks to the Author):

The authors addressed my concerns. It is publishable now.

Response: We are grateful to the reviewer's comments for the agreement of acceptance with this version.

Reviewer #2 (Remarks to the Author):

From my side, I think the authors have sincerely and appropriately addressed all referees' comments and made satisfactory revision with an abundance of supplemented experimental and computational results and discussion. This quality of this updated version is largely improved to meet the standard of Nature Commun. I have no further concern on the manuscript.

Response: We are grateful to the reviewer's comments for the agreement of acceptance with this version.

Reviewer #3 (Remarks to the Author):

Unfortunately, after the revision the manuscript is not suitable for publication in Nature Commun., I recommend rejection.

Response: We are grateful to the reviewer's comments. Based on the comments of the referees, we have addressed these questions and shed light the novelties clearly in our revised manuscript.

I am not convinced about the accuracy of the electrochemical characterizations. There are many discrepancies, for example, authors claim they used 2x3cm (6cm²) Ni foam to prepare the catalyst but in the response letter they mention they have geometrical area of 1cm² catalyst, since the catalyst is merged into solution and both sides of Ni foam is immersed I would expect higher geometric area (if they used whole 6cm² sample and dipped it into electrolyte as they show in the SI, then geometric area will be 12cm²). The response to the comment about ECSA determination raised the following questions: what

was the specific capacitance of Ni, NiMoO_x/NiMoS and all the other catalysts used for finding the ECSA? Were they determined from the plots of current density vs. the scan rate? What is the porosity of this Ni foam, for instance, the BET surface area? I suggest that the authors deposit their catalyst material on a flat surface rather than using a foam and then test the intrinsic activity, because the Ni foam will have much higher surface area than simple geometric area due to its porosity.

Response : We are grateful to the reviewer's comments. The typical catalysts were prepared by use of 2x3 cm² Ni foam with both sides. For the measurement of electrocatalytic performance, 2x3 cm² Ni foam with the catalysts was cut into 1x1 cm² Ni foam. Especially, the effective geometric area of Ni foam with the catalysts is only 1 cm², because 0.5x1 cm² Ni foam with the catalysts as the effective geometric area was merging into the alkali solution while the other part of Ni foam was covered by the epoxy resin.

The electrochemically active surface area (ECSA) gives an estimation of active sites and is proportional to the double-layer capacitance (C_{dl}) which was measured in the non-Faradic region. The slope of capacitive current for cyclic voltammetry plotted versus scan rate gives twice the value of C_{dl}. The C_{dl} values of all catalysts were obtained from the plots of the differences in current density versus the scan rate. For example, the C_{dl} values of NiMoO_x/NiMoS, NiO_x/Ni₃S₂, MoO_x/MoS₂ and NiMoS supported on Ni foam were 23.4 mF cm⁻², 11.1 mF cm⁻², 9.6 mF cm⁻² and 7.9 mF cm⁻², respectively, shown in Figure R1.

To explore the intrinsic activities of the catalysts, cyclic voltammetry curves of NiMoO_x/NiMoS supported on Ni plate were measured for the calculation of C_{dl} values, indicating that the high ECSA value of NiMoO_x/NiMoS supported on Ni plate, as shown in Figure R2.

In order to confirm the determination of the electrochemically active surface area (ECSA) of the catalysts, the following equation (Thomas F. Jaramillo, et al. J. Am. Chem. Soc. 2013, 135, 16977–16987; Thomas F. Jaramillo, et al. Angew. Chem. Int. Ed. 2014, 53, 14433; Thomas F. Jaramillo, et al. J. Am. Chem. Soc. 2015, 137, 4347–4357; Thomas F. Jaramillo, et al. Energy Environ. Sci. 2015, 8, 3022) could be utilized below:

$$ECSA = \frac{C_{dl}}{C_s} \text{cm}_{ECSA}^2$$

where C_{dl} is double-layer capacitance, C_s is specific capacitance.

Figure R1. (abcd) Cyclic voltammetry curves and (e) the difference in current density plotted against scan rate fits to a linear regression for the calculation of electrochemical double-layer capacitance of pristine NiMoS, MoO_x/MoS₂, NiO_x/Ni₃S₂ and NiMoO_x/NiMoS heterostructure supported on Ni foam during HER process.

However, the specific capacitance of the catalysts used for the determination of ECSA is physical character of typical material. Our catalysts are a typical hybrids consisting of NiO_x/Ni₃S₂ and MoO_x/MoS₂, thus it is hard to achieve the accurate specific capacitance of NiMoO_x/NiMoS catalysts. Generally, the specific capacitance for a flat surface is

generally found to be in the range of 20-60 $\mu\text{F cm}^{-2}$ (Thomas F. Jaramillo, et al. J. Am. Chem. Soc. 2013, 135, 16977–16987; Thomas F. Jaramillo, et al. Angew. Chem. Int. Ed. 2014, 53, 14433; Thomas F. Jaramillo, et al. J. Am. Chem. Soc. 2015, 137, 4347–4357; Thomas F. Jaramillo, et al. Energy Environ. Sci. 2015, 8, 3022; Yuliang Li, et al. Nat. Commun. 2019, 10, 2281; Xun Wang, et al. Nat. Commun. 2017, 8, 15377; Xun Wang, et al. Nat. Commun. 2018, 9, 2452). Thomas F. Jaramillo (Stanford University, United States) presented the specific capacitance (C_s) of the sample or the capacitance of atomically smooth planar surface of the material per unit area under identical electrolyte condition. However, specific capacitances have been measured for a variety of metal electrodes in alkaline solutions and typical values reported range between $C_s = 20\text{-}60 \mu\text{F cm}^{-2}$ in KOH solutions. For our estimates of surface area, we use general specific capacitances of $C_s = 40 \mu\text{F cm}^{-2}$ in alkaline solution based on typical reported values. In general, the ECSA estimates to be accurate within about an order of magnitude, and emphasize that the values should be considered only as an approximate guide for comparing electroactive surface area (Thomas F. Jaramillo, et al. J. Am. Chem. Soc. 2013, 135, 16977–16987; Thomas F. Jaramillo, et al. J. Am. Chem. Soc. 2015, 137, 4347–4357). In the following calculations of ECSA, we assume the average value of $40 \mu\text{F cm}^{-2}$ as the specific capacitance of the catalysts in this work. Based on the above equation analysis, the ECSA value of $\text{NiMoO}_x/\text{NiMoS}$ catalyst was roughly calculated to be $83.25 \text{ cm}^2_{\text{ECSA}}$.

Figure R2. (a) Cyclic voltammetry curves of $\text{NiMoO}_x/\text{NiMoS}$ supported on Ni plate. (b) The difference in current density plotted against scan rate fits to a linear regression of $\text{NiMoO}_x/\text{NiMoS}$ supported on Ni plate for the calculation of electrochemical double-layer capacitance.

To further elucidate the difference of NiMoO_x/NiMoS supported on Ni foam and Ni plate, HER polarization curve of NiMoO_x/NiMoS supported on Ni plate were presented in Figure R3 in comparison of NiMoO_x/NiMoS supported on Ni foam, indicating the highest electrocatalytic performance of NiMoO_x/NiMoS supported on Ni foam in this work. Moreover, the high BET surface area, 9.58 m³ g⁻¹ of NiMoO_x/NiMoS was obtained in comparison of Ni foam, as shown in Figure R4, significantly promoting the favorable electrocatalytic water splitting. Based on above-mentioned analysis, the excellent catalytic activity of NiMoO_x/NiMoS supported on Ni foam towards overall water splitting could be rationalized as follows: (i) the synergistic effect of NiMoO_x and NiMoS, results into the abundant exposed active sites and low charge-transfer resistance and (ii) the effective electron transfer configuration through the integrated 3D architecture has been constructed by NiMoO_x/NiMoS supported on Ni foam, providing a large electrochemically active surface area and numerous active sites towards HER and OER. All advantages make the contributes to the enhancement of overall water splitting performance of efficient and stable NiMoO_x/NiMoS electrocatalysts.

Figure R3. HER polarization curves of NiMoO_x/NiMoS supported on Ni foam and Ni plate in 1 M KOH.

Figure R4. N₂ adsorption-desorption isotherms of NiMoO_x/NiMoS and Ni foam.

What was the mass loading of commercial Pt and IrO₂ catalysts onto Ni foam, how do they control this, how do they know the catalyst sticks to the Ni foam? Please measure the ECSA of Pt and IrO₂ catalysts from their respective CVs. Which company they were purchased from? Do they correspond well with their activities reported in the literature?

Response : We are grateful to the reviewer's comments. The mass loading of Pt/C and IrO₂ was same amount of NiMoS_x/NiMoO, 1.6 mg cm⁻². Pt/C and IrO₂ were dispersed in 1ml water/ethanol (v:v, 4:1) solution with 20 μL Nafion solution (5 wt%) by the sonication treatment to form a homogenous ink. The ink was loaded onto the Ni foam. Pt/C (20 wt% Pt) and IrO₂ were purchased from Sigma-Aldrich and Alfa Aesar.

To confirm the existence of commercial Pt/C and IrO₂ catalysts onto Ni foam, the electrocatalytic performance was measured by linear sweep voltammetry (LSV) curves. As shown in Figure R5, the high currents of commercial Pt/C and IrO₂ catalysts onto Ni foam were observed in comparison of pristine Ni foam in 1 M KOH.

Figure R5. (a) HER polarization curves of commercial Pt/C supported on Ni foam in comparison of pristine Ni foam in 1 M KOH. (b) OER polarization curves of commercial IrO₂ supported on Ni foam in comparison of pristine Ni foam in 1 M KOH.

To explore the intrinsic activities of commercial Pt/C and IrO₂ catalysts, linear sweep voltammetry curves of commercial Pt/C and IrO₂ catalysts supported on Ni plates were tested, as shown in Figure R6. Moreover, cyclic voltammetry curves of commercial Pt/C and IrO₂ catalysts supported on Ni plates were measured for the calculation of C_{dl} values, as shown in Figure R7.

In order to confirm the determination of the ECSA of commercial Pt and IrO₂ catalysts, the following equation could also be utilized below:

$$ECSA = \frac{C_{dl}}{C_s} \text{cm}^2_{ECSA}$$

Figure R6. (a) HER polarization curves of commercial Pt/C supported on Ni plate in comparison of pristine Ni plate in 1 M KOH. (b) OER polarization curves of commercial IrO₂ supported on Ni plate in comparison of pristine Ni plate in 1 M KOH.

Figure R7. (ab) Cyclic voltammetry curves of Pt/C and IrO₂ supported on Ni plate. (cd) The differences in current density plotted against scan rate fits to a linear regression from Pt/C and IrO₂ supported on Ni plate for the calculation of electrochemical double-layer capacitances.

Generally, the specific capacitance was found to be in the range of 20~60 $\mu\text{F cm}^{-2}$ and we assume the value of 40 $\mu\text{F cm}^{-2}$ as the specific capacitance of the catalysts in this work. Based on the above analysis, the ECSA values of commercial Pt/C and IrO₂ catalysts supported on Ni plate were roughly calculated to be 187 and 144 $\text{cm}^2_{\text{ECSA}}$.

In order to evaluate the intrinsic electrocatalytic activities of NiMoO_x/NiMoS and commercial Pt/C catalysts supported on Ni plates, the current was normalized to ECSA, as presented in Figure R8. Significantly, NiMoO_x/NiMoS catalyst showed substantially larger HER current density than that of commercial Pt/C catalyst under the same measurement condition.

Figure R8. Polarization curves of NiMoO_x/NiMoS and commercial Pt/C supported on Ni plate with the relative current normalized to ECSA.

Table R1. The overpotential at a typical current density of commercial Pt/C and IrO₂ on Ni foam

Catalysts	Overpotential@10mA cm ⁻²	References
Pt/C	35mV	Nat. Energy 4, 512 (2019).
	23.5mV	Nat. Commun. 11, 4246 (2020).
	35mV	Nat. Commun. 11, 1215 (2020).
	33mV	Nat. Commun. 11, 1278 (2020).
	57mV	Nat. Commun. 11, 1029 (2020).
	77mV	Nat. Commun. 10, 4875 (2019).
	25mV	Nat. Commun. 10, 4977 (2019).
	79mV	Nat. Commun. 10, 2281 (2019).
	30mV	Nat. Commun. 10, 1217 (2019).
	49mV	Nat. Commun. 10, 631 (2019).
	59mV	Nat. Commun. 9, 2551 (2018).
33mV	This work	

IrO ₂	320mV	Nat. Commun. 11, 2701 (2020).
	365mV	Nat. Commun. 11, 1215 (2020).
	406mV	Nat. Commun. 10, 4875 (2019).
	400mV	Nat. Commun. 9, 2452 (2018).
	310mV	Nat. Commun. 9, 2609 (2018).
	320mV	Nat. Commun. 8, 15341 (2017).
	320mV	Adv. Mater. 32, 1906915 (2020)
	294mV	Adv. Mater. 31, 1806672 (2019)
	308mV	Adv. Mater. 30, 1803151 (2018)
	376mV	Angew. Chem. Int. Ed. 59, 14533 (2020)
	377mV	Angew. Chem. Int. Ed. 58, 4923 (2020)
	350mV	This work

Moreover, authors didn't address my comments about the stability tests. For Figures 5d, 6d and 7d, they show the change in current density by holding at certain current density, this simply measures how stable is the potentiostat not the catalyst material. It is meaningful to show how the potential changes over time at constant current or how the current changes at constant potential over time. However, the plot of current density at constant current density over time is meaningless and doesn't provide any information about the catalyst material.

Response : We are grateful to the reviewer's comments. The stability is an important factor for the evaluation of practical electrocatalytic application. At the initial reviewing stage, time-dependent current density curves of NiMoO_x/NiMoS at typical overpotentials have been presented in our manuscript. According to the comment of the reviewer, we have provided the constant potential over time for the current changes in the revised manuscript, while there is no signal in Figures 5d, 6d and 7d. To avoid the misunderstanding these points, we have provided the time-dependent current density curves of NiMoO_x/NiMoS at typical potentials in Figure R2, such as HER activities 100 and 500 mA cm⁻² at 0.089 and 0.174 V vs. RHE over 50 hours, OER activities at 100 and 500 mA cm⁻² with the potentials of 1.455 and 1.508 V vs. RHE over 50 h, typical two-electrode cell at 500 mA cm⁻² with the voltage of 1.75 V over 500 h in 1 M KOH solution at 25 °C. The detailed information has been provided in all Figures and the manuscript.

Figure R9. Time-dependent current density curves of NiMoO_x/NiMoS at typical potentials. (a) HER activities 100 and 500 mA cm⁻² at 0.089 and 0.174 V vs. RHE over 50 hours. (b) OER activities at 100 and 500 mA cm⁻² with the potentials of 1.455 and 1.508 V vs. RHE over 50 h. (c) Typical two-electrode cell at 500 mA cm⁻² at the voltage of 1.75 V over 500 h in 1 M KOH solution at 25 °C.

Moreover, authors claim atomic activation sites, defective structure of the catalyst but provide no experimental result to support this claim. Especially, after it was pointed out by the reviewer that defects on the TEM images are from beam and larger than single

atomic defects. So I would suggest to rename, reconstruct the paper accordingly by avoiding terms such as atomic activation sites and defective structure if they don't provide any experimental evidence.

Response : We are grateful to the reviewer's comments. From the analysis of XPS spectra, the signals at 529.5 and 531.5 eV for O 1s belong to typical metal-oxygen bonds and oxygen vacancies in NiMoO_x/NiMoS heterostructure. With regard to S 2p peaks, the negative shift is observed in NiMoO_x/NiMoS with the increasing temperature of thermal treatment, demonstrating the loss of S and the formation of S vacancies. Thus, the defects have been successfully introduced into NiMoO_x/NiMoS heterostructure, playing a pivot role upon the obvious enhancement of PEC performance. However, it is hard to identify the existence of the defects in NiMoO_x/NiMoS heterostructures according to the analysis of TEM images. As a consequence, it is not necessary to repeatedly highlight the roles of the defects and atomic activation sites. According to the reviewer's comments, we have reconstructed the paper accordingly by avoiding terms such as atomic activation sites and defective structure.

REVIEWERS' COMMENTS

Reviewer #3 (Remarks to the Author):

Authors addressed all my comments/concerns. I have minor comment for the caption of Figure 7d it would be clearer for the readers if authors write: "chronoamperometric test at 1.75V in 1.0 M KOH at 25 C" because they are holding the potential constant at specified potential and record the current over time.

Editorial notes:

Reviewer comments & decisions:

REVIEWER COMMENTS

Reviewer #3 (Remarks to the Author):

Authors addressed all my comments/concerns. I have minor comment for the caption of Figure 7d it would be clearer for the readers if authors write: "chronoamperometric test at 1.75V in 1.0 M KOH at 25 C" because they are holding the potential constant at specified potential and record the current over time.

Responses to the reviewer's comments

Reviewer #3 (Remarks to the Author):

Authors addressed all my comments/concerns. I have minor comment for the caption of Figure 7d it would be clearer for the readers if authors write: "chronoamperometric test at 1.75V in 1.0 M KOH at 25 C" because they are holding the potential constant at specified potential and record the current over time.

Response: We are grateful to the reviewer's comments. The caption of Figure 7d has been modified in the revised manuscript, such as "chronoamperometric test at 1.75 V in 1.0 M KOH at 25 °C".